# Vocal development in a Waddington landscape

**Yayoi Teramoto[1†], Daniel Y Takahashi[1,2*†], Philip Holmes[1,3*], Asif A Ghazanfar[1,2,4*]**

[1]Princeton Neuroscience Institute, Princeton University, Princeton, United States; [2]Department of Psychology, Princeton University, Princeton, United States; [3]Department of Mechanical and Aerospace Engineering and Program in Applied and Computational Mathematics, Princeton University, Princeton, United States; [4]Department of Ecology and Evolutionary Biology, Princeton University, Princeton, United States

**Abstract** Vocal development is the adaptive coordination of the vocal apparatus, muscles, the nervous system, and social interaction. Here, we use a quantitative framework based on optimal control theory and Waddington's landscape metaphor to provide an integrated view of this process. With a biomechanical model of the marmoset monkey vocal apparatus and behavioral developmental data, we show that only the combination of the developing vocal tract, vocal apparatus muscles and nervous system can fully account for the patterns of vocal development. Together, these elements influence the shape of the monkeys' vocal developmental landscape, tilting, rotating or shifting it in different ways. We can thus use this framework to make quantitative predictions regarding how interfering factors or experimental perturbations can change the landscape within a species, or to explain comparative differences in vocal development across species

**\*For correspondence:**
takahashiyd@gmail.com (DYT);
pholmes@math.princeton.edu
(PH); asifg@princeton.edu (AAG)

[†]These authors contributed
equally to this work

**Competing interests:** The
authors declare that no
competing interests exist.

**Reviewing editor:** David
Kleinfeld, University of California,
San Diego, United States

## Introduction

Understanding how behavior changes across development requires a system-level understanding of the interplay among an organism's current behavioral capabilities, its changing body and changing nervous system (*Byrge et al., 2014*). Vocal behavior emerges, at a minimum, from the interactions of the vocal apparatus (the vocal folds, vocal tract and lungs) (*Ghazanfar and Rendall, 2008*), the muscles that control the vocal apparatus, and the nervous system and its interplay with social factors. Development of vocal behavior is the process of adapting a context-dependent stable configuration of these elements so that they work together to produce vocalizations typical of each developmental stage (*Figure 1a*). Yet, there is no theoretical or computational framework in which to understand how the elements of vocal systems come to assemble themselves in this manner during development.

Studies of vocal development typically focus only on one or two of the above elements at any given time. For example, the vocal learning literature emphasizes the role played by imitation and the neural changes that may facilitate this behavior, particularly in songbirds and humans (*Doupe and Kuhl, 1999*). In such considerations, vocal development is not restricted by body structure or motor development, but rather by memory-related constraints and perceptual predispositions. Such a view is incomplete for a number of reasons (*Tchernichovski and Marcus, 2014*; *Vihman, 2015*). For example, young swamp sparrows cannot imitate artificially sped up versions of their species' song, demonstrating muscular constraints on learning (*Podos, 1996*). Along similar

**eLife digest** As infants develop they learn new behaviors and refine existing ones. For example, human infants progress from crying to babbling to producing speech-like sounds. A complex sequence of changes in muscles, the nervous system and in patterns of interactions with other individuals all contribute to these emerging behaviors.

Despite this complexity, most studies of vocal development have only considered single factors in isolation. A study of speech development, for example, might examine how changes in the brain enable infants to imitate sounds. However, that same study will probably ignore how changes in the structure of the vocal cords, or in the behavior of the parents, also promote imitation.

Young marmoset monkeys, like human infants, gradually develop from producing immature cries to adult-like calls. Teramoto, Takahashi et al. built a computational model of this process and compared the model to data from real animals. The first version of the model focused solely on how the marmosets' vocal cords grow, and did not fully reproduce how adult-like calls emerge in real marmosets. Teramoto, Takahashi et al. therefore added factors to the model that simulate improvements in muscle control, learning in the nervous system and in the behavior of other animals. These findings show that, to reflect how adult-like calls emerge in real marmosets, the model needs to include all of these factors.

The model developed by Teramoto, Takahashi et al. may also provide insights into why vocal learning and some other behaviors emerge in some species and not others. It may also be used to predict the consequences of disrupting individual processes in young animals at particular points in time and how such disruptions shape the way an animal develops on its way to adulthood.

lines, in human infants there is not only growth in the vocal apparatus (*Fitch and Giedd, 1999*; *Vorperian et al., 2005*), but developmental changes in vocalization-related motor control (*Green et al., 2000*) and respiration (*Boliek et al., 1996*). Moreover, in songbirds (*West and King, 1988*; *Chen et al., 2016*), bats (*Prat et al., 2015*), marmoset monkeys (*Takahashi et al., 2015*; *Gultekin and Hage, 2017*) and humans (*Kuhl et al., 2003*; *Goldstein et al., 2003*; *Goldstein and Schwade, 2008*), social responses by adults influence vocal development. Given that vocal development consists of a number of moving parts, how can we track and understand how these parts and their relationships change over time to produce mature-sounding vocalizations?

Similar questions, of course, plague all studies of development. Cells, for example, are dynamic entities whose phenotypes change over time. How can we understand the trajectory of a pluripotent stem cell differentiating into a fixed cell type (e.g., a neural stem cell differentiating into a neuron vs. a glial cell vs. remaining a stem cell)? Waddington (*Waddington, 1957*) envisioned this canalized pattern as a ball (the cell's state) rolling down a surface with hills and valleys to seek the lowest points in an *epigenetic landscape*. At watersheds, the valleys branch so that the ball takes one of two available paths and thus establishes its identity at that particular time. Recently, this metaphor for cellular development was given a formal quantitative theoretical framework (*Wang et al., 2011*). Cells have states defined by the expression patterns of interacting genes. These states correspond to different basins of attraction in a probability landscape; cell differentiation proceeds as movement from one basin to another (*Wang et al., 2011*). All forms of biological development – including vocal development – are probabilistic like this cell fate example (*Gottlieb, 2007*). Vocalizations also go through different states as they transition from immature to mature forms (*Kent and Murray, 1982*; *Tchernichovski et al., 2001*; *Scheiner et al., 2002*; *Lipkind et al., 2013*; *Takahashi et al., 2015*; *Zhang and Ghazanfar, 2016*) (e.g., for marmoset monkeys, see *Figure 1b*). These states are defined by the probabilistic relationship between the vocal apparatus, muscle control, neural activity and social context (*Figure 1a*).

In the current study, our goal is to generate an integrated landscape framework for vocal development that incorporates these elements and their interactions over time. To do so, we will use marmoset monkey vocal development, which shares numerous parallels with human vocal development (*Takahashi et al., 2015*; *Zhang and Ghazanfar, 2016*; *Takahashi et al., 2016*; *Ghazanfar and Zhang, 2016*). First, infant marmoset monkey call acoustics change during development

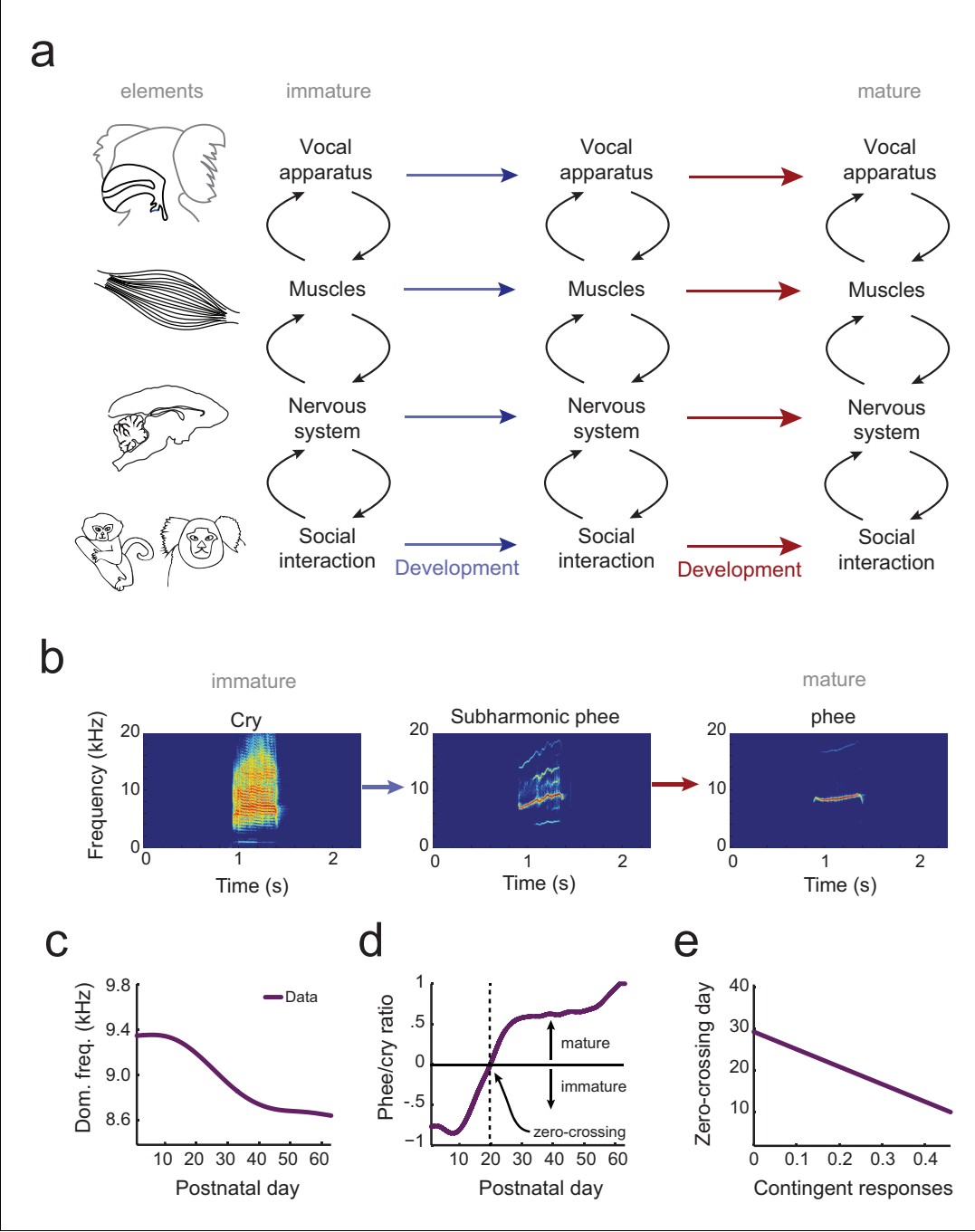

**Figure 1.** The elements of vocal development and their interactions. (**a**) Vocal development is the result of changes in, and interactions among, the vocal apparatus, muscles, nervous system, and social context. (**b**) Infant marmosets produce mostly immature calls (cries and subharmonics) during early postnatal days which are replaced by more adult-like calls (phees) during development. (**c**) Changes in vocal acoustics during development include a lowering of the dominant frequency. Purple curve shows a cubic spline fit to the data. (**d**) Change in the proportion of mature calls compared to immature calls (the phee/cry ratio). Purple curve shows a cubic spline fit to the data. The zero-crossing day is the postnatal day in which the number of cries and phees are the same, marking the transition from mature to immature vocalization. (**e**) Relationship between the probability of parental contingent responses and the zero-crossing day. Purple line shows the linear regression fit to the data.

(*Pistorio et al., 2006*; *Takahashi et al., 2015*; *Zhang and Ghazanfar, 2016*) (*Figure 1c*). Second, these changes in acoustics reflect the transition from an initial mixture of immature and mature vocal sounds to adult-like vocalizations (*Takahashi et al., 2015*, *2016*; *Zhang and Ghazanfar, 2016*) (*Figure 1d*). Third, as in humans (*Goldstein et al., 2003*; *Kuhl et al., 2003*; *Goldstein and Schwade, 2008*), the timing of this transition is influenced by contingent parental vocal feedback (*Takahashi et al., 2015*, *2016*; *Gultekin and Hage, 2017*) (*Figure 1e*). Finally, after taking into account their rapid growth relative to humans (*de Castro Leão et al., 2009*), changes in the developmental trajectory of marmoset vocal behaviors occur at the same life history stages (*Takahashi et al., 2015*, *2016*; *Zhang and Ghazanfar, 2016*; *Ghazanfar and Zhang, 2016*). Using an extensive longitudinal vocal behavioral dataset from marmoset infants (*Takahashi et al., 2015*, *2016*; *Zhang and Ghazanfar, 2016*), collected under two controlled contexts (brief social isolation (*undirected context*) and vocal interactions with a parent (*directed context*)), we applied optimal control principles to formulate and test the predictions of a landscape framework for vocal development. This landscape shows how changes in the vocal apparatus, muscles, nervous system, and social interaction together shape the vocal developmental trajectory of an infant (*Figure 1a*).

## Overview of approach

In our study, the specific vocal behavior under investigation is the production of mature contact ('phee') calls. Adult marmoset monkeys produce these vocalizations when alone and out of sight of others (undirected context) (*Borjon et al., 2016*). If another marmoset is within earshot, then the pair will begin taking turns exchanging these calls (directed context) (*Takahashi et al., 2013*). Very young infants are only gradually able to produce mature sounding contact calls (*Takahashi et al., 2015*; *Zhang and Ghazanfar, 2016*), and contingent vocal interactions with parents appears to accelerate this process (*Takahashi et al., 2015*, *2016*). Here, we use optimal control theory to construct a Waddington-like developmental landscape to model this process.

Optimal control approaches have long been used in studies of motor behaviors and their application requires four specifications: (1) well-defined behaviors, (2) a biomechanical model of the system, (3) a cost function, and (4) an optimization criterion that describes the probabilities of those behaviors (*Wolpert and Landy, 2012*). The theory posits that the probability of producing a specific motor action can be calculated by knowing the cost that a given behavior demands (*Wolpert and Landy, 2012*). If the cost to produce an action is high, that action should be less probable than another whose cost is lower. In the current study, the four specifications are the following: (1) Immature and mature contact calls are the behaviors; (2) The biomechanical model is one established for songbird vocalizations (*Amador and Mindlin, 2008*; *Perl et al., 2011*; *Amador et al., 2013*) that we have shown is also appropriate for marmoset monkeys (*Takahashi et al., 2015*); (3) The cost function is the amount of effort required to produce contact calls; and (4) The optimization criterion is the maximum entropy principle. Maximizing the entropy allows us to identify the probability distribution that is most consistent with the cost function and makes the fewest assumptions. In essence, the goal of our study is to understand how each of the elements of vocal behavior – the vocal apparatus, muscles, nervous system, social interaction – modifies this cost function over postnatal days.

The overall pattern of vocal (contact call) development consists of a change in dominant frequency, a rapid transition from immature to mature calls, and a correlation between the amount of parental feedback and the rate of this transition (*Figure 1c–e*). We will use the optimal control approach to take the following inferential steps in order to explain this pattern of vocal development (*Figure 2*). First, we will use the biomechanical model to simulate growth of the vocal apparatus (specifically, the vocal tract length) (*Figure 2a,b*). We will then fit the model's parameters so that it can reproduce the dominant frequency changes observed in marmoset monkey vocal development (*Figures 1c* and *2c*). Second, we will test whether these changes in the vocal tract length can also account for the rapid transition from immature to mature contact calls (phee/cry ratio; *Figure 1d*). To do this, we combine the cost function (*Figure 2d*) with the optimization criterion which together generate a probability distribution for the production of immature and mature calls (*Figure 2e*). Third, the prediction is either falsified or supported by comparing the model-based phee/cry ratio with the real marmoset phee/cry ratio (*Figure 2f*). If the prediction is falsified, we add a new element to the cost function (e.g, change in muscle control) which changes it shape and thus changes the probability distribution of call types produced in the emerging landscape. We then repeat the inferential steps using the vocalization data, cost function, and optimal control theory (*Figure 2g*). To

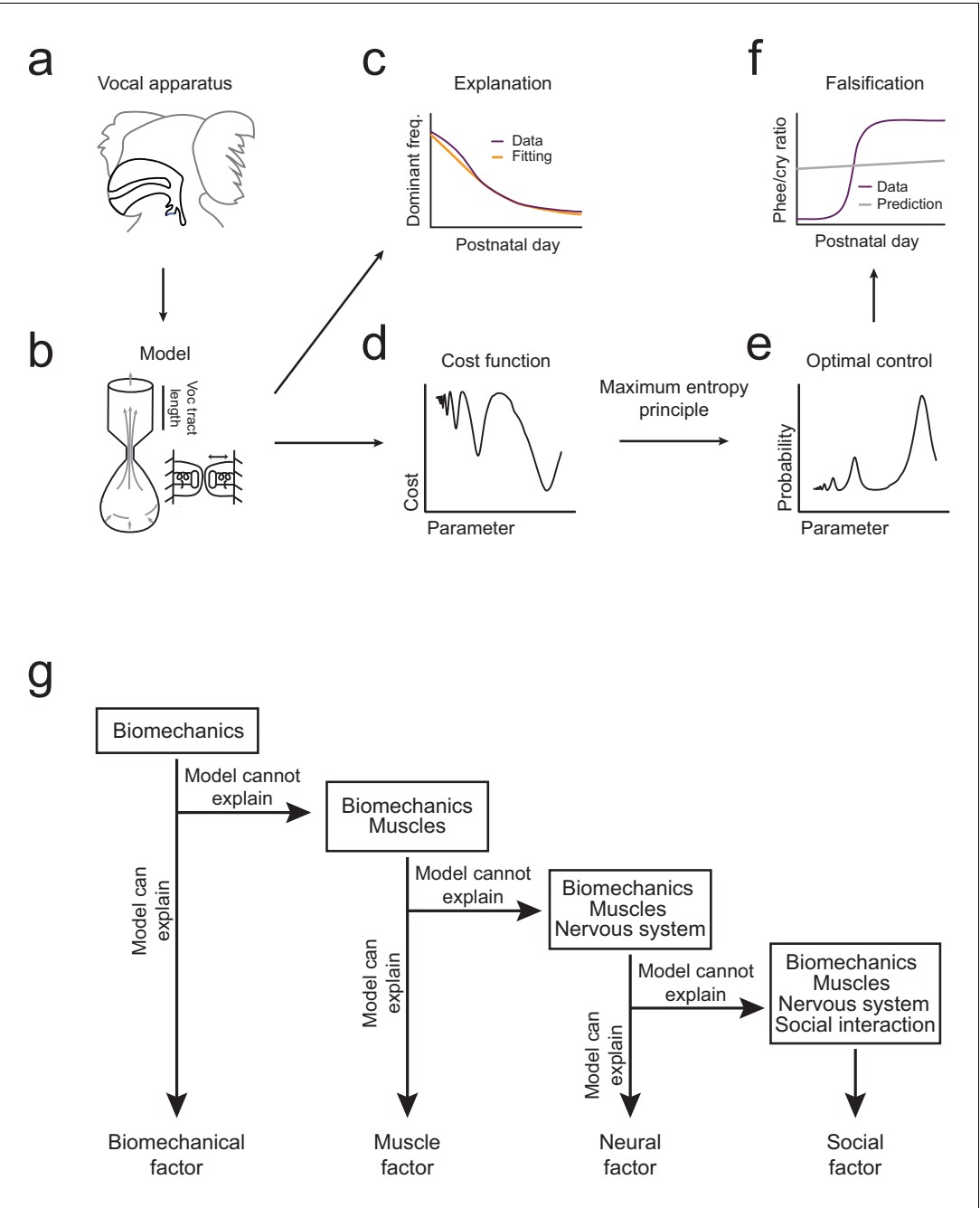

**Figure 2.** Illustration of the inferential process used in the study. (**a,b**) A biomechanical model is made of the infant marmoset monkey vocal apparatus. (**c**) The model is used to simulate how the growth of the vocal tract lowers the dominant frequency of calls. Model data (yellow line) can be fitted to the real data (purple line). (**d,e**) Optimal control theory is used to generate a cost function for producing different call types and the maximum entropy principle is used to calculate a probability distribution. (**f**) Using the probability distribution, we can calculate the phee/cry ratios produced by the simulated vocal tract growth (gray line) and compare with the real marmoset phee/cry ratio data (purple line). (**g**) The contributions of other individual elements (see *Figure 1a*) are gradually added to the framework using a sequential inferential approach together with mathematical modeling.

distinguish when a statement is about the model or about the real data, we will always indicate the corresponding model parameter when discussing the model. With the intent to make the main message of the article as clear as possible, we postpone most of the mathematical content to Materials

and methods and the Appendix. The reader interested in the mathematical aspects of the modeling will find callouts in relevant places of the main text.

In what follows, we first present the biomechanical model of the marmoset vocal apparatus as this serves as the foundation of our optimal control approach. We then present our findings related to the growth of the vocal tract and the successive additions of muscle, nervous system and social interaction to the developmental landscape.

## Results

### A biomechanical model of the marmoset monkey vocal apparatus

Establishing a biomechanical model for the vocalizations produced by developing marmoset monkeys is required for the optimal control approach. Briefly, we use a model that is a second order ordinary differential equation with two possible time-varying parameters: $\alpha(t)$, representing the air pressure produced by the lungs and $\beta(t)$, representing vocal fold tension (*Figure 3a*). Different values of $\alpha$ and $\beta$ generate different combinations of air pressure and laryngeal tension, resulting in distinct acoustic signals. The third parameter $\gamma$ is a fixed inverse time scale that sets the upper frequency range of glottal (vocal fold) oscillations. The glottal air flow ($P_{glottal}$) is then filtered by the vocal tract to produce the final vocal output ($P_{sound}$). The vocal tract is modeled as a cylinder in which the filtering property depends on its length $L$ and reflection coefficient $r$ (*Figure 3a*). Details of the model are described in Materials and methods: The vocal fold model and From vocal vibrations to calls, *Equations (14–17)*; parameter values are given in *Table 1* and further mathematical details appear in the Appendix.

By varying the air pressure $\alpha$ and vocal fold tension $\beta$, the model produces immature and mature contact calls (cries, subharmonic-phees and phees) with nearly identical acoustic features to those produced by infant marmosets (*Figure 3b–d*); it can also simulate sequences of calls (*Figure 3e*). Respiration $\alpha$ and vocal fold tension $\beta$ can change in time to produce the different call types. To obtain the results in *Figure 3b–e*, $\alpha$ and $\beta$ were varied as increasing and/or decreasing linear ramps. *Figure 3f* shows the parameter regions that result in each call type. Lower respiratory power $\alpha$ and vocal fold tension $\beta$ produce cries, whereas higher values produce phees. When $\alpha$ and $\beta$ are small (gray region, *Figure 3f*) there is no vocal production. Physiological respiratory data support the predictions of the model (*Takahashi et al., 2015*).

By varying the parameters, the fundamental frequencies and amplitude of vocal sounds can be changed. Higher fundamental frequencies are obtained when the air pressure $\alpha$ and/or the laryngeal muscle tension $\beta$ increases (*Monsen et al., 1978*; *Hollien, 2014*). Consistent with this, *Figure 3g* shows that the model has isofrequency (same frequency) lines for glottal airflow that increase with higher air pressure $\alpha$ and/or muscle tension $\beta$. Vocal amplitude is mainly controlled by the air pressure (*Sundberg et al., 1993*), which the model expresses as nearly vertical iso-amplitude (same amplitude) curves in *Figure 3h*. The glottal air flow is then filtered by the resonant vocal tract. The gain $g$ is measured as the ratio between the amplitudes of vocal output (after vocal tract resonance) and of glottal air flow (before vocal tract resonance) ($g(\alpha, \beta) = \max_t P_{sound}(t) / \max_t P_{glottal}(t)$). Glottal air pressures that oscillate at the resonance frequencies produce higher gains than those that do not (*Ghazanfar and Rendall, 2008*). *Figure 3i* shows the effect on the gain produced by different values of air pressure and muscle tension. The highest gains are obtained for glottal airflow at approximately 9–10 kHz (*Figure 3g,i*).

### Growth of the vocal tract contributes to lower dominant frequency

In humans and other primates, vocal development includes a lowering of the dominant frequency of calls (*Hammerschmidt et al., 2000*, *2001*; *Kent and Murray, 1982*; *Scheiner et al., 2002*; *Pistorio et al., 2006*; *Takahashi et al., 2015*) (*Figure 1c*). Such changes in frequency in early vocal acoustics are typically associated with increases in the size of the vocal folds: as they get bigger they naturally oscillate more slowly, producing lower frequency sounds. Some early vocalizations are also noisy (see the cry in *Figure 1b*). Noisiness in vocal acoustic features in general are typically associated with instabilities in the vocal fold movements (*Kent and Murray, 1982*; *Fitch et al., 2002*; *Tokuda et al., 2002*). Our initial modeling study of the biomechanics of marmoset monkey vocal development revealed that, unexpectedly, the vocal tract may additionally play an important role in

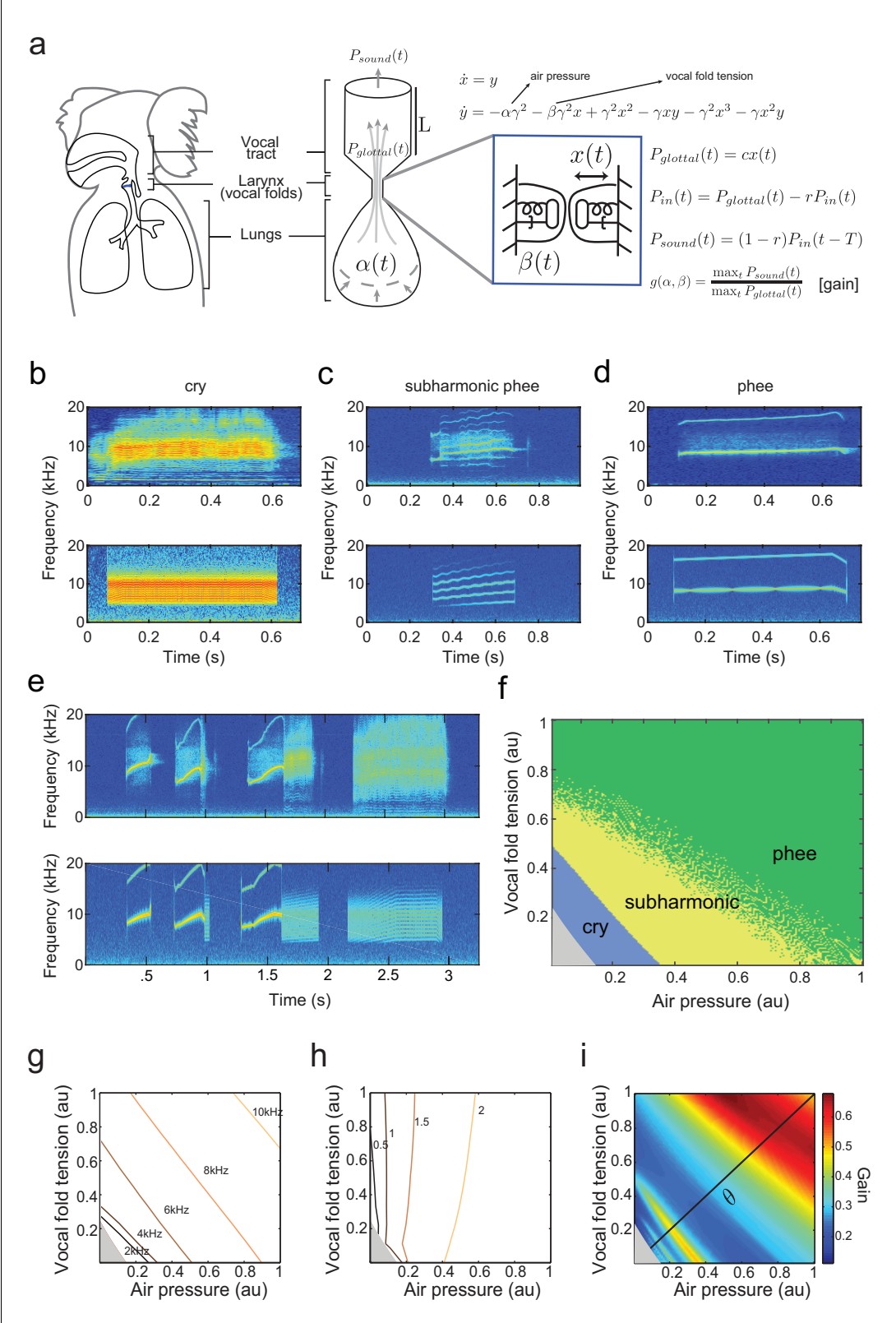

**Figure 3.** A biomechanical model of marmoset vocal apparatus. (**a**) Representation of the biomechanical model of the vocal production apparatus. In our one-mass model $x(t), y(t)$ are displacement and velocity of vocal folds; nondimensional lung air pressure, vocal fold tension and overall inverse timescale are represented by parameters $\alpha(t), \beta(t)$ and $\gamma$. Glottal exit air flow $P_{glottal}$ is filtered by the vocal tract, modeled as a cylinder of length $L$ with reflection coefficient $r$ at the mouth, to produce vocal output $P_{sound}$. $T/2 = L/c_{sound}$ is the one way travel time with sound speed $c_{sound}$. (**b–d**) Examples of

**Figure 3 continued**

real infant calls (top) and model simulation of the same calls (bottom). (**e**) Example of a sequence of infant calls (top) and model simulation (bottom). (**f**) Different values of air pressure and vocal fold tension produce distinct types of calls. Gray region represents parameter values that do not produce vocalization (i.e., self-sustained oscillation). (**g**) Isofrequency curves. Lines show air pressure and vocal fold tension values that produce glottal air flow that oscillates at the same frequencies; parameters in the gray region do not produce self-sustained oscillations. (**h**) Iso-amplitude curves. Lines show air pressure and vocal fold tension values that produce glottal air flow with same amplitudes. (**i**) Plot showing gains: the ratios between sound produced after the resonance (vocal output) and before the resonance (glottal air flow); warmer colors indicate higher ratios. The diagonal line ($\alpha = \beta$) is parametrized by $\theta$. au = arbitrary units.

generating the acoustic features present in both immature and mature vocalizations (*Takahashi et al., 2015*). Thus, in this study, we explore the role of vocal tract growth on shaping the developmental landscape.

When an animal's body size increases during development, so does the length of its vocal tract (*Fitch and Giedd, 1999*). Since longer vocal tracts have lower main and subharmonic resonance frequencies $f_0, f_0/2, f_0/3$, etc., we expect the resonance frequency to decrease over development. To test this, we fitted the developmental change in dominant frequency observed in the undirected context data (*Figure 4a*) and estimated the developmental change in this feature due to the changing length of the vocal tract $L$ (*Figure 4b*). As expected, the increase in $L$ and the associated changes in resonance frequencies during development can explain the observed reduction in the dominant frequency of vocalizations. Thus, the change in dominant frequency is a developmental feature that can be associated with changes in vocal tract length. Having established that, we can now use optimal control theory to determine if vocal tract length $L$ can also explain other features of the infant marmoset vocal development. In particular, we will examine if the change in $L$ can explain the rapid transition from producing mostly immature vocalizations like cries and subharmonic-phees to mostly adult-like contact phee calls (*Figure 1d*) (*Takahashi et al., 2015*; *Zhang and Ghazanfar, 2016*). To do so, we will need to calculate the probability to produce immature and mature calls. Optimal control theory will allow us to do this, but first we must define an ethologically relevent 'cost' of producing vocalizations.

Based on what we know about the ethology of infant marmoset monkeys, there are benefits to producing vocalizations with higher gains (i.e., vocalizations that are louder, longer and more tonal) (*Figure 4c*). Marmoset infant cries, subharmonic-phees, and phees are produced when they are separated from the parents (*Takahashi et al., 2015*). These vocalizations are louder compared to other infant calls and result in parents approaching the infant, and so are considered contact calls (*Newman, 1985*). However, infant marmoset calls that are more tonal (or 'phee' like; [*Figure 1b*, right panel]) are more likely to elicit parental responses (*Takahashi et al., 2016*). Hence, we model the cost of producing a call at different air pressure and vocal fold tension as inversely related to the gain $g(\alpha,\beta) = \max_t P_{sound}(t) / \max_t P_{glottal}(t)$. We can therefore write the cost to produce a vocalization with a given air pressure ($\alpha$) and vocal fold tension ($\beta$) as

$$C(\alpha,\beta) = -\log g(\alpha,\beta), \tag{1}$$

where $\alpha,\beta \in [0,1.1]$ remain in the region of viable calls (see *Figure 3f*). The higher the gain for this function, the lower the cost. The logarithm is used to make the unit of gain proportional to decibels (dB).

To simplify our analysis and allow visualization, in what follows we will consider only the diagonal section $\alpha = \beta$ of the parameter space, labeled $\theta$, that passes through the region of cries, subharmonic-phees, and phees. Other choices of $\alpha$ and $\beta$ that include these three calls yield similar results. The cost function *Equation (1)* becomes

$$C(\theta) = -\log g(\theta), \tag{2}$$

where $\theta \in [0,1.1]$. *Figure 4d* illustrates our first 'landscape': the cost function with troughs indicating where glottal air pressure oscillates at the vocal tract's resonance frequency and subharmonics $f_0, f_0/2$, etc. This cost function describes one section of the developmental landscape related to respiration and vocal fold tension.

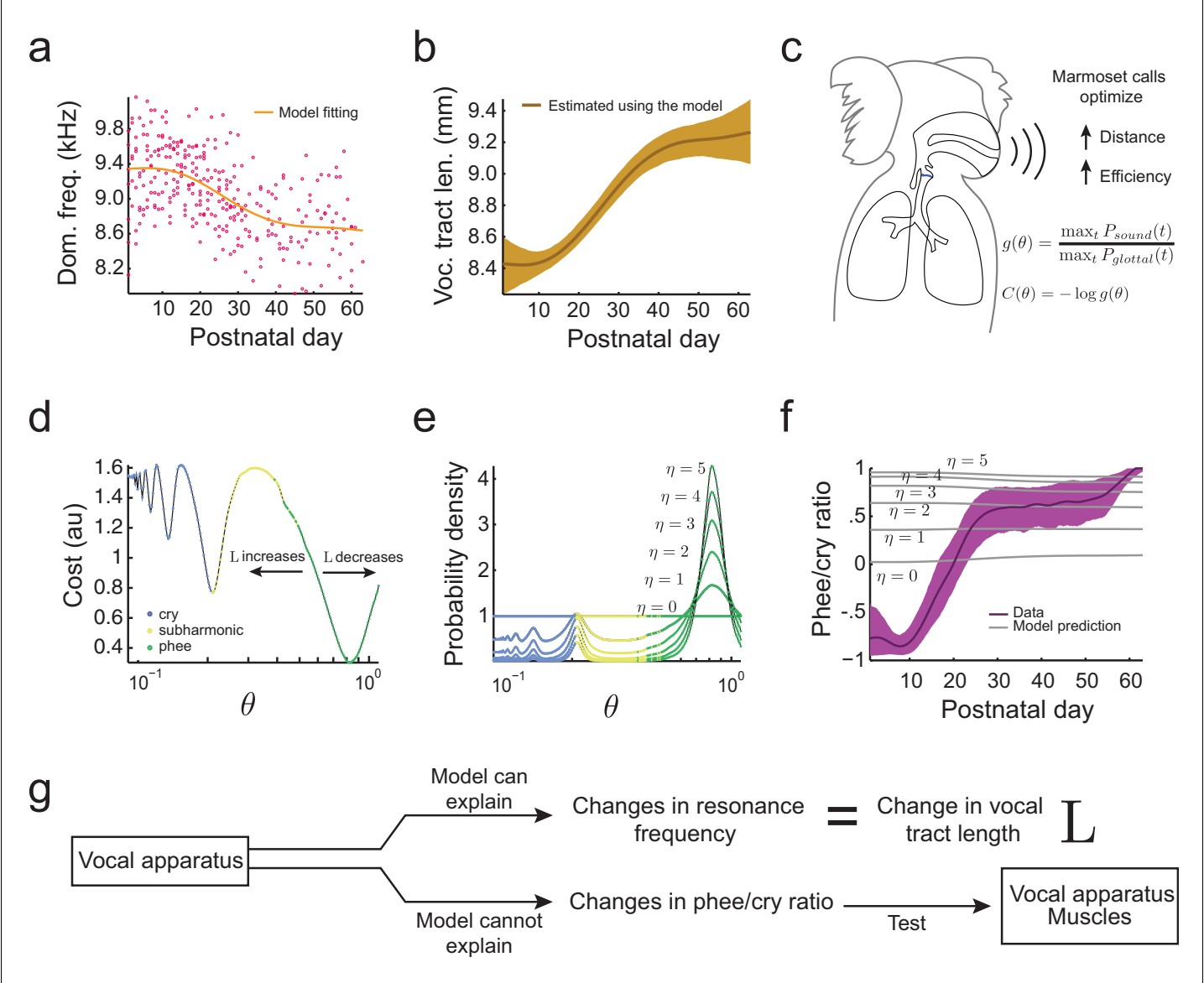

**Figure 4.** Growth of the vocal tract. (**a**) Change in dominant frequency of infant marmoset calls during development. Yellow curve shows the value of resonant frequency fitted by the biomechanical model. Red dots are the mean dominant frequency of each postnatal day for all 10 infants ($n = 301$ sessions). (**b**) Vocal tract length estimated by the model assuming a closed-closed cylindrical tube (brown curve); shaded region indicates 95% confidence interval. (**c**) Infant marmosets produce calls that maximize distance and efficiency. Therefore, the cost $C(\theta)$ of producing a call is inversely related to the gain $g(\theta)$. (**d**) Cost function to produce calls at different air pressure and vocal fold tension values ($\theta$). Blue, yellow, and green dots indicate parameter regions for cry, subharmonic-phee, and phee production, respectively. Minimal cost is achieved for phees, which have glottal air flow oscillating at the natural frequency of the vocal cavity; $\theta$-axis is in log-scale. (**e**) Probability density to produce calls at different $\theta$ values; color code is the same as in (**d**). Increasing $\eta$ concentrates probability in the parameter range that produces phees. (**f**) Population and model phee/cry ratios. Purple line is the population value of phee/cry ratio for the real marmoset infant data; shaded region indicates 95% confidence interval ($n = 195$ sessions). Gray lines indicate phee/cry ratios predicted by the model for different values of $\eta$. (**g**) Growth (lengthening) of the vocal tract can explain the lowering of the dominant frequency, but not the transition from cries to phees.

We can now describe the effect of developmental changes in vocal tract length $L$ on the shape of this landscape. An increase in $L$ causes a decrease in the location of the troughs with respect to $\theta$, and vice-versa (*Figure 4d*). The different color regions indicate the different types of calls produced by the model for a given $\theta$. Minimal cost is obtained when the infant produces mature contact phee calls because the frequency of glottal oscillations match $f_0$. Given the cost function, we want to

predict the probability that the infant will produce a call with a given air pressure and vocal fold tension. This is achieved by using the maximum entropy principle, via application of the softmax action selection rule (*Jaynes, 1982*; *Wilson et al., 2014*). This will give the probability of producing different calls that is consistent with the cost function and makes the fewest possible assumptions (see Materials and methods: Softmax action selection rule for details). This rule implies that the probability to produce a call with a given $\theta$ is proportional to the exponential of the negative of the cost:

$$\mathrm{Prob}(\theta) = \exp(-\eta C(\theta))/Z. \tag{3}$$

Here $\eta$ is a non-negative parameter that controls the concentration of the probability distribution and that can be estimated from the data. $Z = \int \exp(-\eta C(\theta))\mathrm{d}\theta$ is the normalizing constant such that the total probability is one. *Figure 4e* shows that increasing $\eta$ increases the probability to produce phees. When $\eta$ is zero, all parameter values are equally likely and we obtain the minimum possible proportion of phees.

Now we can ask a key question. Is a developmental landscape that only incorporates changes in vocal tract growth sufficient to explain not only lowering of the dominant frequency (*Figure 1c*), but also the other features of marmoset monkey vocal development? If so, then it should be able to explain the rapid transition from immature to mature calls during development (*Figure 1d*; [*Takahashi et al., 2015*]). To test this hypothesis, we calculated the phee/cry ratio, defined as

$$\mathrm{phee/cry\ ratio} = \frac{\mathrm{Prob(phee)} - \mathrm{Prob(cry)}}{\mathrm{Prob(phee)} + \mathrm{Prob(cry)}}, \tag{4}$$

for the data and the model. Using the model, we can calculate the probability to produce a specific type of call by integrating the probability density for the air pressure and vocal fold tension that produce each type of call. Specifically, if $A_{cry}$ is the set of parameters $\theta$ for which the model produces cries (*Figure 4e*, blue region), we have

$$\mathrm{Prob(cry)} = \int_{A_{cry}} \mathrm{Prob}(\theta)\mathrm{d}\theta. \tag{5}$$

Similarly, if $A_{phee}$ is the set of parameters for which the model produces phees (*Figure 4e*, green region), we have

$$\mathrm{Prob(phee)} = \int_{A_{phee}} \mathrm{Prob}(\theta)\mathrm{d}\theta. \tag{6}$$

*Figure 4f* (gray lines) shows that during development, changes in vocal tract length $L$ have only a small influence on the phee/cry ratio and increasing $\eta$ only increases the probability of phees. But the phee/cry ratio in the marmoset data is negative for early postnatal days, showing more cries, and exhibiting a fast transition to mostly phee production after $20-30$ postnatal days. Therefore, there are no values of $\eta$ and $L$ that can fit the data and the cost function that includes only the change in vocal tract length cannot predict the cries-to-phees transition observed in development (*Figure 4f,g*). In other words, the changes in the position of troughs in the landscape due to vocal tract length increases are insufficient to explain other features of vocal development beyond lowering of the dominant frequency. Therefore, we will next consider the development of muscular control in the vocal apparatus.

## Development of both vocal tract and muscle control accounts for the rapid transition from immature to mature vocalizations

Laryngeal and respiratory muscle size, strength, and dynamics significantly change through postnatal development in humans (*Moore, 2004*; *Sasaki, 2006*). We expect the control of respiratory and laryngeal muscles to change similarly during development in marmoset monkeys. Based on this assumption, one possibility is that the larger proportion of cries that occurs in the early postnatal period is due to very young infants having difficulty producing higher air pressures and vocal fold tensions required to generate mature (phee) calls (*Figure 3f*). Producing higher values requires stronger respiratory and laryngeal muscles and greater coordination (*Takahashi et al., 2015*). Our

aim, therefore, is to estimate a new cost function and hence developmental landscape based on both vocal tract growth *and* the development of muscular control. We will model the cost of muscular control by modeling the required muscular effort as $\lambda\theta$: a linear function of $\theta$ with a parameter $\lambda$ whose values define how steep is the change in muscular effort for larger values of air pressure and vocal fold tension. *Figure 5a* shows the muscular effort for different values of $\theta$ and $\lambda$. In this second function, the total cost to produce a call for a given value of $\theta$ is the sum of the cost of the vocal tract change *Equation (2)* and muscular effort:

$$C(\theta) = -\log g(\theta) + \lambda\theta, \tag{7}$$

for $\theta \in [0, 1.1]$. *Figure 5b* shows this cost function for different values of $\theta$ and $\lambda$. Higher values of $\lambda$ increase the cost to produce phees (green) more rapidly than the cost to produce cries (blue). Therefore, the effect of adding $\lambda\theta$ to the cost function is to rotate the developmental landscape counterclockwise, increasing the cost of producing phees.

Using the maximum entropy principle as before (softmax action selection rule *Equation (3)*), we can calculate the probability to produce calls for a given $\theta$. As expected from the effect of $\lambda$ on the cost function (*Figure 5b*), *Figure 5c* shows that higher values of $\lambda$ imply a lower probability to produce phees and higher probability to produce cries. This indicates that the developmental transition from cries to phees can be a consequence of a decrease in $\lambda$ (i.e., an increase in muscular control) during development. To test this possibility, we fitted the phee/cry ratio data using the cost function *Equation (7)* (*Figure 5d*). The fit follows the phee/cry ratio curve obtained from the directed context data obtained from infant marmosets (*Figure 4f*). *Figure 5e* shows the values of $\lambda$ estimated by applying the model to these real data. As expected, we find that $\lambda$ decreases during development (i.e., muscular control increases).

Thus, a two-element developmental landscape that includes vocal tract growth and the development of muscle control of the vocal apparatus can account for two key features of vocal development: lowering of the dominant frequency as calls become more mature and the rapid transition from early immature calls to mature ones. Our next question is whether this two-element landscape can also explain individual variability in the timing of the rapid transition. This timing is represented by the zero-crossing day (*Figure 1d,e*) when the number of immature and mature calls produced is equal (*Takahashi et al., 2015*). Our prior work demonstrated that the individual timing of the zero-crossing day appears to depend upon the number of contingent responses provided by parents when they hear the infant's contact calls (*Takahashi et al., 2015*). Thus, to answer this question, we calculated the correlation between the zero-crossing day and the probability of contingent parental responses to infant calls (*Takahashi et al., 2015*). We observe that there are clear correlations between the amount of parental feedback and the rate of the cry-to-phee transition (*Figure 5f*, purple line) but these cannot be explained by the cost function that only includes the elements of vocal tract growth and muscular control improvements (*Figure 5f*, gray line). Therefore, an additional factor is needed, one that can control the vocal apparatus and is influenced by social feedback – the nervous system (*Figure 5g*).

## Learning in the nervous system facilitated by social feedback accelerates the individual rate of vocal development

As in songbirds (*West and King, 1988*; *Chen et al., 2016*) and humans (*Kuhl et al., 2003*; *Goldstein et al., 2003*; *Goldstein and Schwade, 2008*), contingent parental responses appear to influence vocal development in marmoset monkeys (*Takahashi et al., 2015*, *2016*). The timing of transition from a cry-dominated early developmental period to phee-dominated later period is correlated with the amount of contingent parental vocal feedback that each infant receives (*Figure 1e*) (*Takahashi et al., 2015*). Contingent parental responses are those that are produced within 5 s of an infant call. Infants that receive a higher proportion of contingent parental calls exhibit earlier transitions from cries to phees. This, of course, is social feedback-based reinforcement learning mediated by large-scale networks in the nervous system (*Syal and Finlay, 2011*). Given that increasing muscular control (i.e., decreasing $\lambda$) increases the phee/cry ratio, we hypothesize that the change in the nervous system driven by social feedback affects the daily rate at which $\lambda$ decreases during development. In light of this, the amount of change in $\lambda$ would be proportional to the amount of parental feedback that the infant receives: a larger proportion of parental feedback will decrease $\lambda$ by a

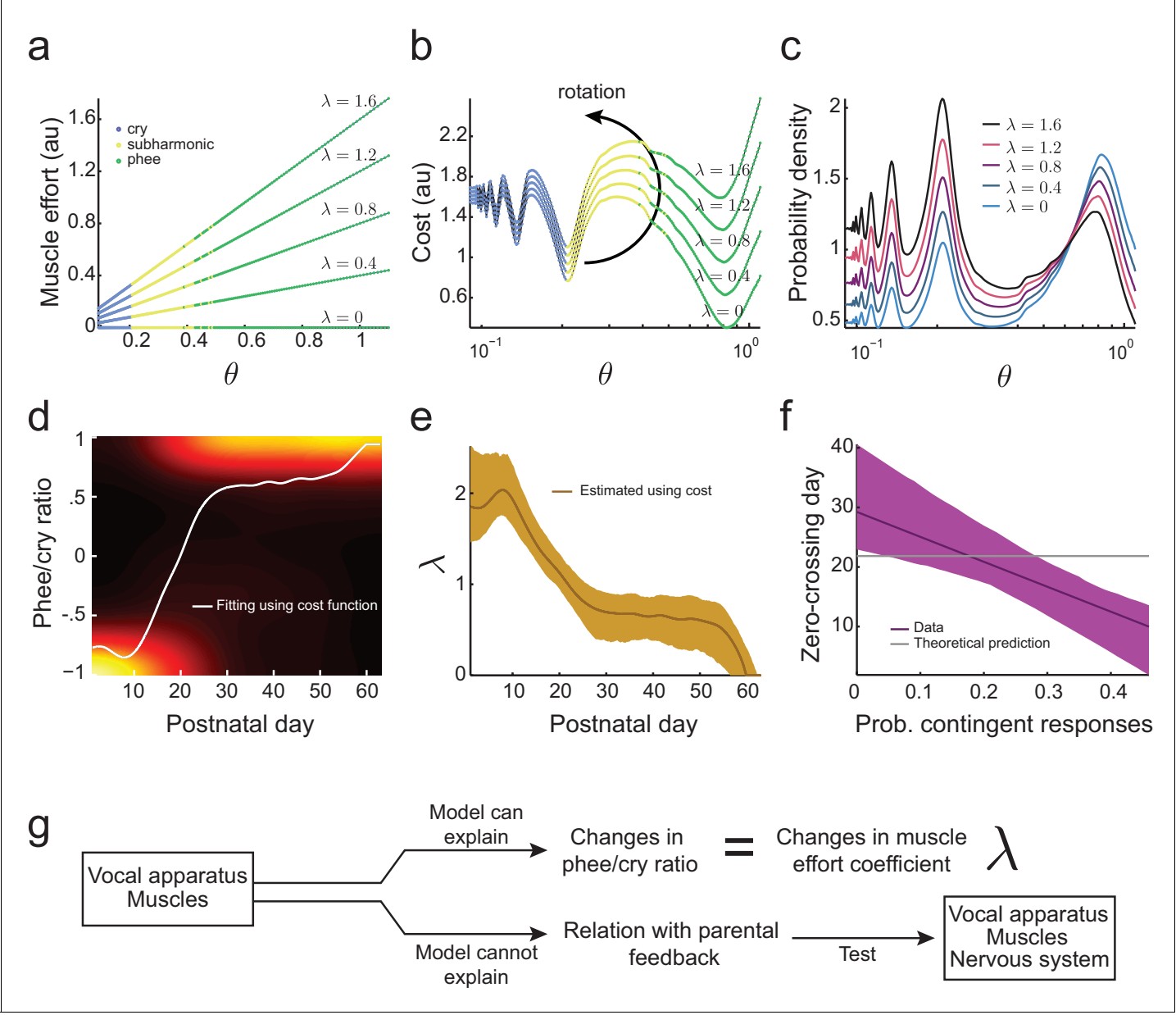

**Figure 5.** Development of muscular control in the vocal apparatus. (a) Muscular control necessary to produce different air pressure and vocal fold tension; higher values of $\lambda$ imply a greater effort to produce given air pressure and vocal fold tension. Blue, yellow, and green dots indicate parameter regions for cry, subharmonic-phee, and phee production, respectively. (b) Cost functions for different values of $\lambda$. (c) Probability to produce calls at different air pressure and vocal fold tension. For higher values of $\lambda$, probability to produce phee diminishes and the probability to produce cries increases. (d) Phee/cry ratio fitted by the model (white curve). Colors indicate the probability density of the phee/cry ratio for the marmoset population ($n = 195$ sessions); warmer colors indicate higher probability densities. (e) Estimated muscle effort coefficient ($\lambda$) during development (brown curve); shaded region indicates 95% confidence interval ($n = 195$ sessions). (f) Relationships between the probability of contingent parental responses and zero-crossing day for real data (purple line) and the model (gray line); shaded region indicates 95% confidence interval ($n = 10$ infants). (g) Changes in muscular control can explain the population change in the phee/cry ratio, but not the social feedback-influenced the individual timing of this transition.

larger amount. Therefore, we propose the following relationship between the value of $\lambda$ as a function of time, $\lambda_t$, indexed by postnatal day, and the average proportion of contingent parental feedback, represented by $F$:

$$\lambda_t = \lambda_{t-1} - \kappa F - \delta. \tag{8}$$

Here $\kappa$ is a parameter that models the effect of learning and can be calculated from the data. $\delta$ models the neuromuscular development that is independent of contingent parental calls. Like human infant babbling (**Koopmans-van Beinum et al., 2001**), infant marmosets will eventually produce adult-like calls with little or no parental feedback (**Takahashi et al., 2015**; **Gultekin and Hage, 2017**). Thus, the daily change in $\lambda$ decomposes into two parts: one ($\kappa F$) that depends on parental feedback and another ($\delta$) that is independent of such feedback. **Equation (8)** implies that $\lambda$ decreases linearly with $t$:

$$\lambda_t = \lambda_0 - (\kappa F + \delta)t, \tag{9}$$

where $\lambda_0$ is the starting value at postnatal day 0. The new cost function for each postnatal day which includes vocal tract growth, muscular control *and* nervous system development is

$$C_t(\theta) = -\log g_t(\theta) + \lambda_0 \theta - \delta t \theta - \kappa F t \theta, \tag{10}$$

where the subscript $t$ indicates dependence on time. **Equation (10)** derives from **Equation (7)** with $\lambda$ replaced by $\lambda_t = \lambda_0 - (\kappa F + \delta)$ from **Equation (9)**.

**Figure 6a** shows the effect of different proportions of contingent parental calls on the development $\lambda$ of as predicted by this cost function. If there is no parental vocal feedback ($F = 0$), e.g., the infant is deaf or raised in social isolation, $\lambda$ still decreases, but at a slower rate determined by $\delta$ (black line). **Figure 6b** shows that the proportion of parental feedback is negatively correlated to the timing of transition from cries to phees. Therefore, learning in the developing nervous system facilitated by social feedback tilts the developmental landscape, so that the transition from cries to phees happens sooner and faster. **Figure 6c** (blue dots) shows the relationship between the proportion of contingent parental calls and the zero-crossing day in the data and the same relationship fitted using the cost function **Equation (10)** (yellow curve, see Materials and methods: The full cost function and more parameter choices for further details). The fitting shows that the relationship between the proportion of contingent parental responses and the rate of transition from cries to phees can be explained by the development of the nervous system facilitated by parental feedback. Nevertheless, this cost function does not explain why parents produce different amounts of contingent calls. Therefore, we have to consider how the social interaction with parents may depend on other variables of infant vocal development (**Figure 6d**).

## Infant growth rate does not influence the probability of contingent responses from parents

The interactions between parents and an infant are predictive of overall health, quality of attachment and the subsequent communication skills of the child. Unhealthy infants who do not vocalize a lot tend to be fed and held less by mothers, and are slowed in their speech development and thus adversely affect the probability of interactions with parents (**Zeskind, 2013**; **Lester, 1985**). Differences in such vocal output can be related to differences in growth (**Zeskind and Lester, 1981**). Therefore, one hypothesis is that infant marmoset monkeys with faster growth rates call more and, as a result, receive more contingent feedback from parents which would accelerate the transition from immature to more mature calls. If this is true, then the higher frequency of parental feedback should be a consequence of parents responding to healthier, more vocal infants. If the hypothesis is falsified, it would suggest that the direct effect of parental feedback is to change the infant's developing nervous system, thereby affecting the rate of this vocal transition independently of overall growth rates.

To model these relationships, let $W$ and $N$ respectively be the weight change over development (a measure of growth) and the call rate of the infant marmosets. We can write the frequency of parental feedback $F$ as a simple linear function:

$$F = b_0 + b_1 W + b_2 N + \epsilon, \tag{11}$$

where $\epsilon$ is noise independent of $W$ and $N$, $b_0$ is the intercept, and $b_1$, $b_2$ are coefficients relating $W$ and $N$ to $F$. If $b_1$ or $b_2$ is different from zero, we have evidence of an indirect effect. To test this hypothesis, we fitted **Equation (11)** to the infant marmoset vocalization data collected in the directed context. We find that no coefficient $b_i$ is significantly different from zero ($n = 10$, $b_0 = 0.083$,

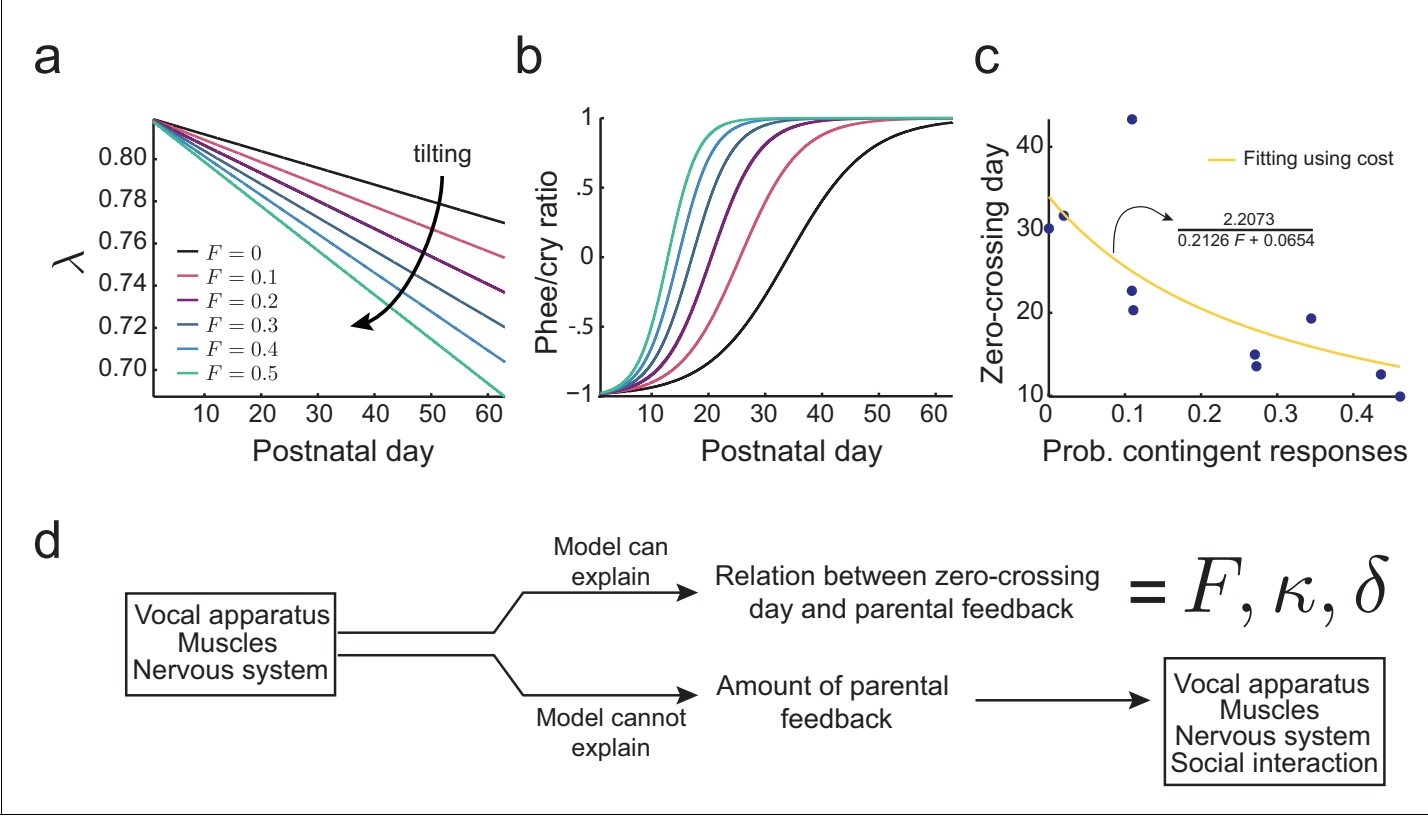

**Figure 6.** Learning in the developing nervous system. (a) Developmental change of $\lambda$ for different values of the probability of contingent parental response, $F$, with constant learning parameter $\kappa = 0.2126$ (see Materials and methods: The full cost function and more parameter choices). Higher values of parental feedback cause faster decay of $\lambda$. (b) Predicted phee/cry ratios for different values of the probability of contingent parental responses. Higher values of parental feedback cause earlier and faster transitions from cries to phees. Color code is the same as in (a). (c) Relationship between the probability of contingent parental response and zero-crossing day; blue dots represent real data ($n = 10$ infants) and yellow line is the model fit. (d) Changes in the nervous system can explain the relation between the rate of transition from cries to phees and the probability of contingent parental feedback, but not the amount of parental feedback.

$p = 0.675$, $b_1 = 0.290$, $p = 0.361$, $b_2 = -0.051$ $p = 0.678$). We also tested whether $W$ and $N$ are separately correlated to $F$ (*Figure 7a,b*). Again, both correlations are not significantly different from zero (respectively, $p = 0.378$ and $0.896$). This corroborates the alternative hypothesis that parental feedback has a direct effect on the infant nervous system that cannot be accounted for by the growth or call rates of infants.

## A dynamic and integrated Waddington landscape for vocal development

What makes an infant marmoset transition from immature to mature-sounding vocalizations? By combining the influences of the developing vocal tract, muscles of the vocal apparatus and the nervous system, we can now present an integrated landscape of vocal development in the manner envisioned by Waddington (*Waddington, 1957*). *Figure 8a* summarizes the relationships between these different elements of vocal production and the corresponding changes in vocal development. In our framework, these elements define the dynamics of the cost function, *i.e.*, the shape of the developmental landscape. *Figure 8b* illustrates the landscape plotted over $(\theta, t)$-space. Its interpretation is as follows: (1) Development of vocal tract length $L$ changes the resonance frequency by shifting the troughs/valleys of the landscape represented by the shape of $g_t(\theta)$ (*Figure 8c*); (2) neuromuscular maturation increases the probability to produce phees by reducing the cost function by an amount $\delta\theta$ per day, *i.e.* rotating the landscape from one postnatal day to the next (*Figure 8d*); and (3) nervous system development driven by social feedback further increases the probability to produce

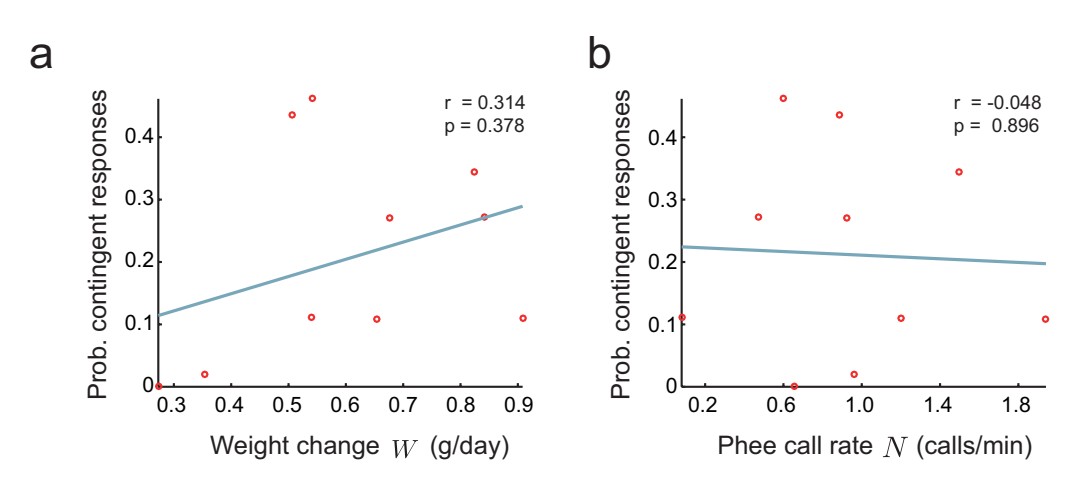

**Figure 7.** Relationship between parental feedback and infant growth. (**a**) Relationship between rate of infant weight change $W$ and the probability of parental responses $F$. Red circles represent data ($n = 10$ infants). Line indicates linear fit; $r =$ Pearson correlation. (**b**) Relationship between rate of infant phee call production $N$ and probability of parental responses $F$; plot convention as in (**a**).

phees by tilting the entire landscape by an amount that is the product of the learning rate $\kappa$ and the proportion of parental feedback $F$ (***Figure 8e***).

To better visualize the dynamics of the landscape as it applies to an individual marmoset infant's vocal development, we can associate a diffusion process to it (***Video 1***). The video shows the states of a particle driven by the gradient of the cost function $C_t(\theta)$ of ***Equation (10)*** and white noise, on 11 postnatal days separated by 6-day intervals. The position of the particle indicates the call types produced on that postnatal day and the amount of time spent producing each call. Much like the basins of attraction proposed for cell differentiation (***Wang et al., 2011***), the deeper the valley, the longer the diffusion process spends in it each day. As time elapses, the cost function $C_t(\theta)$ deforms so that the probability of observing cries decreases and phees become more likely, with a zero crossing day in the third or fourth week, depending on the individual. See Materials and methods: Softmax action selection rule for more information.

## Discussion

Vocal development is a systems-level phenomenon. Its understanding requires the analysis of changes in the vocal apparatus, associated muscles, the nervous system and social interactions. Each of these elements modifies the others and itself over time (***Thelen and Smith, 2006***; ***Byrge et al., 2014***). Using data from developing marmoset monkeys and optimal control theory, we generated a systematic and quantitative inferential framework based on Waddington's developmental landscape metaphor (***Waddington, 1957***). We used it to account for three features of marmoset monkey contact call development: the lowering of the dominant frequency, the rapid transition from producing mostly immature to mostly mature calls, and the influence of social feedback on the timing of this transition (***Takahashi et al., 2015***).

We showed that the change in the dominant frequency of infant vocalizations can be explained by developmental increases in the length of the vocal tract. However, vocal tract growth could not account for the timing of the transition from immature vocalizations (cries), which are abundant in early postnatal days, to mature vocalizations (phee calls) which exemplify later periods. This transition can, however, be explained by including the development of muscular control. This suggests that immature respiratory and laryngeal muscles do not allow the infant marmoset to produce adult-like phees: calls that demand greater effort and/or coordination (***Takahashi et al., 2015***; ***Zhang and Ghazanfar, 2016***). The development of the vocal tract and muscular control, however, could not explain how parental feedback influences the timing of the transition from immature to mature vocalizations. Including a learning component mediated by the nervous system allowed us to infer a

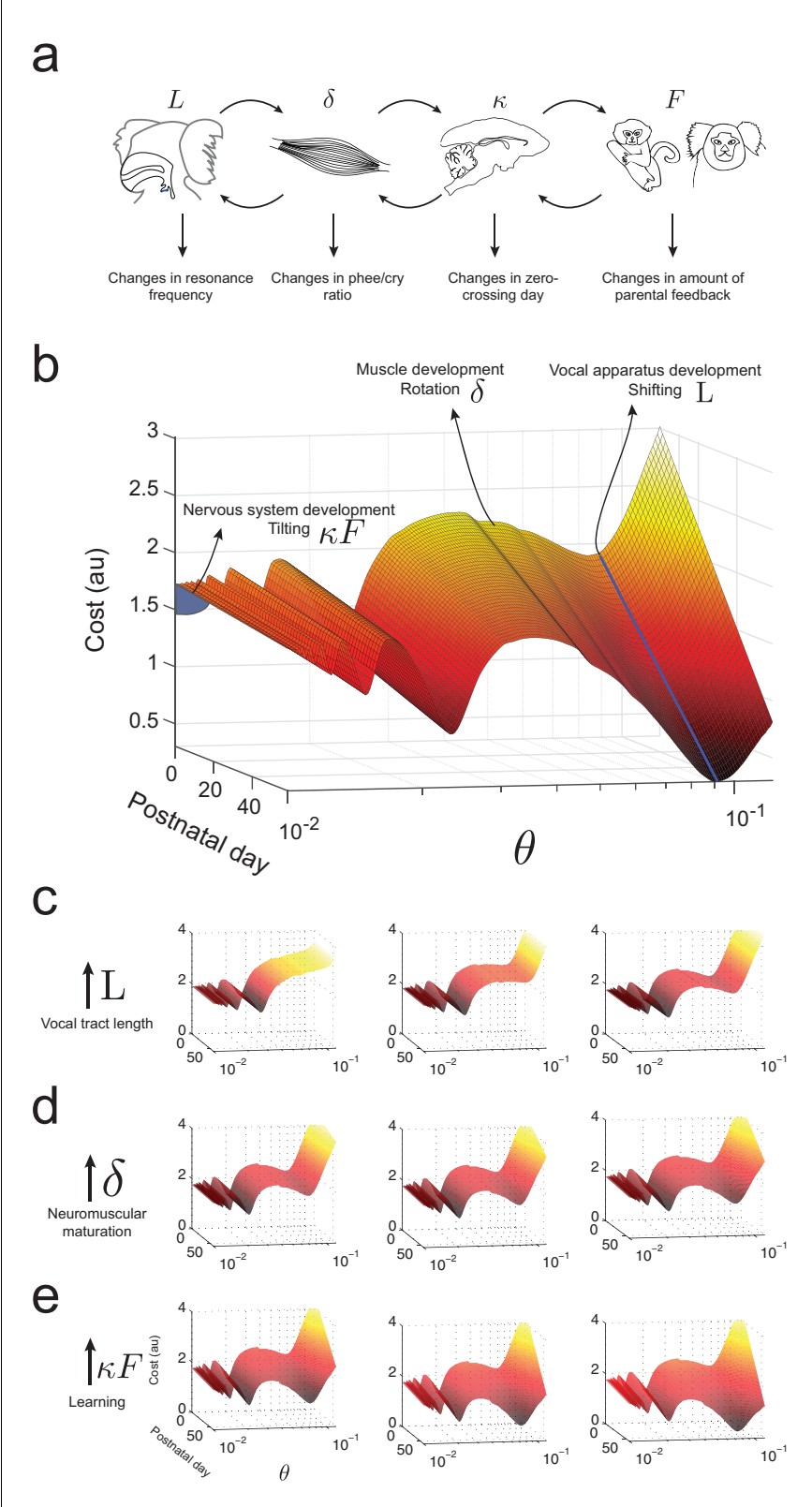

**Figure 8.** Waddington landscape for vocal development. (a) Developmental changes associated with each vocal component: vocal tract length $L$, neuromuscular maturation $\delta$, learning rate $\kappa$, and parental feedback $F$. (b) Different components of vocal behavior change distinct features of the developmental landscape. Similar colors indicate regions with the same cost values; darker colors indicate lower costs. The blue solid line shows the natural frequency of the vocal tract, which depends upon its length $L$. Neuromuscular maturation parameter $\delta$ changes the shape of the landscape.
*Figure 8 continued on next page*

*Figure 8 continued*

The nervous system, influenced by parental feedback $\kappa F$, changes the slope of the landscape, speeding up development as $t$ increases; $\theta$-axis represents values in logarithmic scale. (**c**) Change in landscape as vocal tract length $L$ increases for fixed $\delta, \kappa F$ (left to right). (**d**) Change in landscape as neuromuscular maturation $\delta$ increases for fixed $L, \kappa F$ (left to right). (**e**) Change in landscape as learning rate $\kappa$ times amount of parental feedback $F$ increases for fixed $L, \delta$ (left to right). See *Table 2* for parameter values.

relationship between contingent parental vocal responses and the rate of vocal maturation in individual infants. Thus, incorporating vocal tract growth, increased muscular control and learning-related changes in the nervous system into a single landscape allowed us to see how these elements interact over time and influence the trajectory of vocal development. This underscores the fact that neural networks do not function in isolation; they must typically process sensory data and communicate with muscles to create appropriate behaviors. The resulting coupling with physiological systems both enables and constrains the behaviors that such neural circuits can produce (*Chiel and Beer, 1997*; *Tytell et al., 2011*).

## Vocal biomechanics of developing marmoset monkeys

The key to our optimal control-based elaboration of the vocal development landscape was the biomechanical model for vocal production. The model was originally developed to describe bird song production (*Amador and Mindlin, 2008*; *Perl et al., 2011*; *Amador et al., 2013*) and then adapted to model infant marmoset vocal production (*Takahashi et al., 2015*). The main advantage of the model is its ability to produce all infant marmoset calls by varying only two parameters: air pressure and vocal fold tension; continuous changes in these parameters can produce spectrally distinct cries, subharmonic-phees, and phees. These are sufficiently distinct that they were previously considered to be different types of calls (*Pistorio et al., 2006*; *Bezerra and Souto, 2008*).

The ability of our biomechanical model to generate such acoustic diversity contrasts with previous models. For example, the origin of cries in nonhuman primates has been attributed to turbulent or chaotic dynamics of the vocal folds (*Fitch et al., 2002*), perhaps as a consequence of vocal fold asymmetry (*Herzel, 1993*) and/or source-vocal tract interactions (*Hatzikirou et al., 2006*). Our model produces cries simply through a mismatch between the low frequency periodic glottal air flow and the higher frequency resonance of the infant's upper vocal tract; no chaotic dynamics occurs. The primary difference between cries and phee calls is that the frequency of glottal oscillations is lower in the former (see *Figure 9* (left)). This result provides direct biomechanical support for the hypothesis that cries are the scaffolding for vocal maturation in both marmosets (*Takahashi et al., 2015*) and humans (*Kent and Murray, 1982*).

## Vocal development as the transformation of a cost function

In our study, vocal development is understood as a transformation of the cost function through time as a consequence of changes in the vocal apparatus, muscles, nervous system, and social interaction. To calculate the probability that an infant marmoset produces cries, subharmonic-phees, or phees, we first defined the cost of producing a call with a given air pressure and vocal fold tension. We then calculated the probability of producing each type of call using the maximum entropy principle. The idea of a cost that changes in time to describe development goes back at least to Waddington's epigenetic landscape metaphor (*Waddington, 1957*), but in Waddington's formulation the metaphorical landscape is static and the paths that phenotypical differentiation might take are genetically determined. Modern perspectives using Waddington's landscape metaphor (including the current study) think of development as probabilistic and allow the landscape to change shape over time (*Thelen and Smith, 2006*; *Wang et al., 2011*; *Ferrell, 2012*; *Sasahara et al., 2015*). For example, Sasahara et al. investigated the development of rhythmic structure in the songs of Bengalese finches using a landscape perspective. They showed that rhythm development exhibits branching and new trajectories along which early, simple vocalizations developed into diverse note types followed by specific silent gaps. The trajectory patterns differed considerably among individual birds, but rhythm proficiency progressed exponentially in all birds (*Sasahara et al., 2015*).

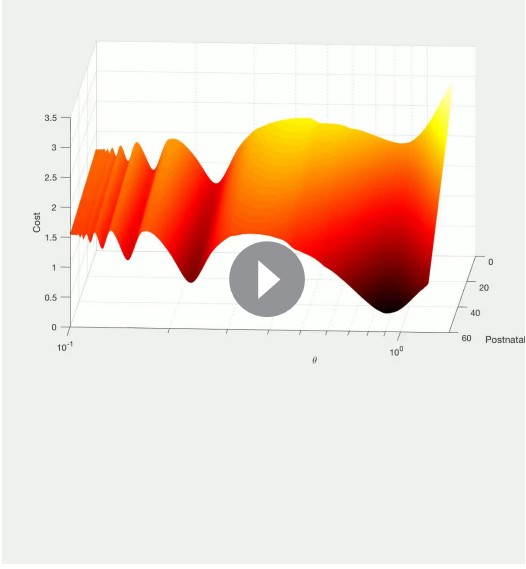

**Video 1.** Animation showing a typical realization of a diffusion process with cost function $C$ as described in Materials and methods: Softmax action selection rule. The particle travels through a developmental landscape that changes its shape due to changes in vocal apparatus, muscle strength, nervous system, and social interaction. The particle's location represents the behavior of a marmoset infant. In early postnatal days, it stays mostly in the parameter region ($\theta$) producing immature calls, whereas in later postnatal days, it stays mostly in the region producing more mature calls. Diffusion dynamics are shown at intervals of six days. Lower left panel shows the numbers of cries and phees produced in each simulated postnatal day; lower right panel shows the phee/cry ratio for the same postnatal days.

## Some caveats: Selection of vocal elements, other behaviors, shape of trajectories and sequential order

In our probabilistic landscape, we inferred the role of vocal tract growth, muscular control and the influence of social feedback on nervous system development. This allowed us to explain – in an integrative manner – the role these elements together play in the transformation of immature to mature contact calls in developing marmosets. We used these somewhat generic elements to most clearly illustrate (in our view) the developmental phenomena, as there is no prior study of this kind. However, a more detailed landscape could certainly be generated by at least three means. First, more elements could be added. For example, lowering of the dominant frequency may also be due to growth-related increases in the size of the vocal folds (*Hammerschmidt et al., 2000*, *2001*), but we only considered the vocal tract. Similarly, 'muscular control' and 'nervous system' in our landscape could be more specifically represented by separating the development of individual muscles and neural connections, respectively, related to vocal apparatus control.

Second, other infant behaviors may act as scaffolding or otherwise constrain or facilitate vocal development (*Iverson, 2010*). In the case of infant marmosets, the ability to self-monitor (and thus to take turns vocalizing with parents) matures in an experience-independent manner at the same time as they transform immature contact calls into mature versions (*Takahashi et al., 2016*). The current study did not incorporate how such changes in self-monitoring could also shape the developmental landscape for this vocal transformation.

Third, we made assumptions about the developmental trajectory of the elements. For example, we assumed that the development of muscular control and learning in the nervous system were linear processes. This simplification has the benefit of making clear the main phenomena in our framework, but more precise data on the developmental trajectories of muscles or learning-related neuronal activity would provide more accurate predictions. Our framework is general enough to incorporate such details for a deeper understanding. For example, if the linear functions can be replaced by more realistic, perhaps non-linear, functions relating air pressure, vocal fold tension and muscular control, they could be incorporated.

Finally, one part of our inferential sequence was that increased muscular control was due to learning-related changes in the nervous system via social reinforcement. An alternative inferential sequence could have been adopted. For example, improvements in muscular control independent of learning could have resulted in more mature-sounding infant calls and thereby increased the rate of parental vocal feedback. This would lead to a different explanation of the correlation between the rate of transition from cries to phees and the amount of parental feedback. We did not test this possibility in our inferential sequence because this hypothesis would be valid only if the change in social interaction were incorporated in the model before changes in the nervous system. The behavioral data do not support this alternative sequence of events: parental call rate, and strength of the

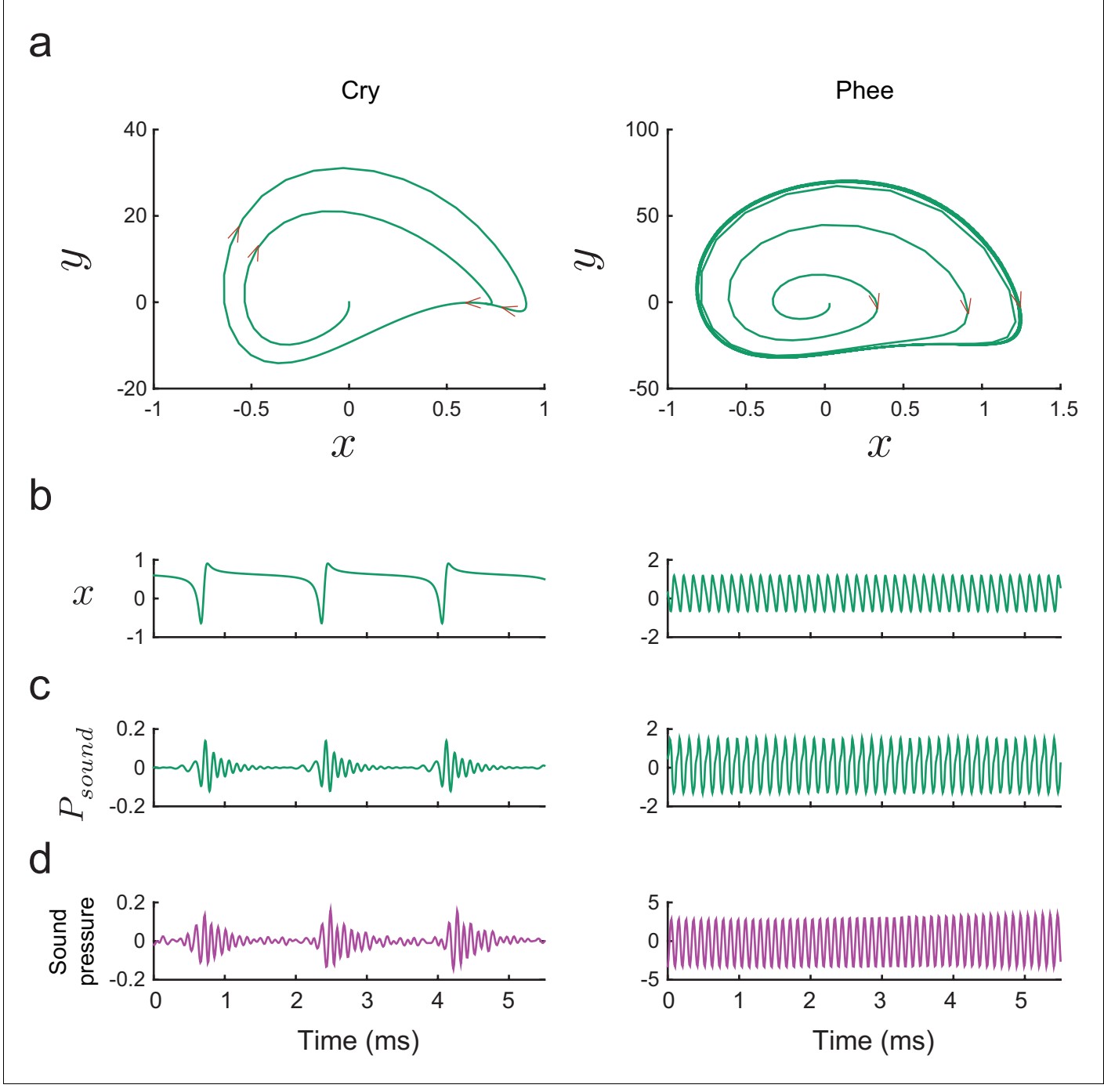

**Figure 9.** Producing marmoset cries and phees with the model. (a) Trajectories of $x$ plotted vs. $y$ for *Equation (14)* for a cry (left) and a phee (right). Parameter values $(\alpha, \beta) = (0.09364, 0.088)$ for cry and $(0.151, 0.895)$ for phee respectively. (b) Glottal air flows $P_{glottal}$ produced by the model and (c) vocalizations $P_{sound}$ produced after resonance in the vocal tract for a cry and a phee. (d) Cry and phee waveforms for calls recorded from infant marmosets; compare with model waveforms shown in (c). Note different vertical scales on left and right columns, indicating that phees are substantially louder than cries.

dynamic interaction between infants and parents, remain constant throughout development (*Takahashi et al., 2015*, *2016*). Thus, a change in social interaction driven by muscle development (before learning-related neural changes) cannot explain the relationship between parental feedback and rate of transition to more mature calls.

## Applications of the vocal development landscape

An integrative understanding of vocal development is important for a variety of reasons, because while we know that many communication disorders originate in problems early in life, we lack any clear grasp of the initial problems. By the time a child is diagnosed with a disorder, the symptoms represent a build up of earlier developmental events. For example, the early vocalizations of infants elicit attention, care and vocal responses from parents (*Lester, 1985*; *Zeskind, 2013*). Infants who do not vocalize much tend to be fed and held less by mothers, and are slowed in their vocal development. The lack of adequate early vocal output by infants may be due to many factors, including abnormal growth of the vocal apparatus, weak laryngeal and respiratory muscles, and/or problems related to nervous system function, such as arousal dysregulation or deficits in motor control and learning.

Understanding the mechanisms for human communication, and how it may go awry, requires the use of model animals that naturally exhibit at least a subset of similar communicative behaviors. The early vocal development of marmosets shares a number of parallels with prelinguistic vocal development in humans (*Ghazanfar and Zhang, 2016*), perhaps due to convergent evolution of a cooperative breeding strategy (*Borjon and Ghazanfar, 2014*; *Ghazanfar and Takahashi, 2017*). Moreover, we are gaining knowledge of the genetics of this species (*Harris et al., 2014*) and, more specifically, the sensorimotor physiology related to its vocal production (*Eliades and Wang, 2008*; *Miller et al., 2015*; *Zhang and Ghazanfar, 2016*; *Borjon et al., 2016*; *Roy et al., 2016*). Recent innovations establishing genetically-modified marmosets (*Sasaki et al., 2009*; *Okano et al., 2016*) will allow for any number of experimental routes needed to gain novel insights into vocal development. The landscape framework in the current study could be used to make quantitative predictions on the effects of genetic or other types of experimental manipulations. For example, the landscape framework combined with genetic engineering could be used to make predictions regarding the influences of communication- or connectivity-related genes expressed during neuroembryological development in marmosets (*Matsunaga et al., 2013*; *Kato et al., 2014*).

Naturally, marmosets do not share with humans every aspect of postnatal vocal development. Songbirds, for example, are much better suited to investigate the shared mechanistic basis for more sophisticated forms of vocal learning (*Lipkind et al., 2013*), though such learning occurs at different life-history stages. The vocal development landscape may be used to illuminate why there are species differences in both the degree to which vocalizations can be learned and the life history-timing of such learning. For example, vocal development data from songbirds and humans could be used to generate landscapes for comparison with the marmoset landscape. Closely related species which differ radically in their vocal behavior could also be compared in this manner. For instance, the landscapes of New World squirrel monkeys, whose vocalizations change very little during development (*Hammerschmidt et al., 2001*), could be quantitatively compared to each other and with the marmoset landscape. Similarly, evolutionary insights could be gained by comparing vocal development landscapes of the white-rumped munia and its domesticated counterpart, the Bengalese finch, whose song behaviors and biologies differ considerably (*Katahira et al., 2013*; *Suzuki et al., 2014*). Moreover, as the evolution of a phenotype in essence defines its developmental trajectory, providing the developmental parameters for different species could illuminate how changes in their respective landscapes lead to similarities or differences in their adult vocal behaviors.

Overall, we believe that the integrated systems view provided by the vocal development landscape not only eschews the incorrect view that there are privileged levels of understanding behavior and its development (*Noble, 2012*; *Krakauer et al., 2017*), but also enables us to make predictions regarding how natural or experimental perturbations (e.g., changes in social feedback, weakening of muscles, disruptions of neural circuits, genetic engineering, etc.) will affect the development of vocal behavior, and why species differ in their capacity to learn communication signals.

## Materials and methods

### Subjects

All experiments were approved by the Princeton University Institutional Animal Care and Use Committee. The data analyzed in this work is a subset of the dataset that was previously published (*Takahashi et al., 2015*) and can be found at http://science.sciencemag.org/content/suppl/2015/08/

13/349.6249.734.DC1. The subjects used in the study were 10 infants and six adults (three male-female pairs, >2 years old), captive common marmosets (Callithrix jacchus) housed at Princeton University. The colony room is maintained at a temperature of approximately 27°C and 50–60% relative humidity, with a 12L:12D light cycle. Marmosets live in family groups; all were born in captivity. They had ad libitum access to water and were fed daily with standard commercial chow supplemented with fruits and vegetables. Additional treats (peanuts, cereal, fruits and marshmallows) were used prior to each session to transfer the animals from their home-cage into a transfer cage.

## Experimental procedures

The vocalizations of marmoset monkey infants were recorded starting on the first postnatal day in two different contexts: undirected (i.e., social isolation) and directed (with auditory, but not visual, contact with their mother or father). The details of the full experiments were described previously (*Takahashi et al., 2015*). Here, the experimental procedures are described in brief for the convenience of the reader. Early in life, infants are always carried by a parent. Thus, the parent carrying the infant(s) was first lured from the home cage into a transfer cage using treats. The infant marmoset was then gently separated from the adult and taken to the experiment room where it was placed in a second transfer cage on a flat piece of foam. The testing corner was counterbalanced across sessions. A speaker was placed at a third corner equidistant from both testing corners and pink noise (amplitude decaying inversely proportional to frequency) was broadcast at 45 dB (at 0.88 m from speaker) in order to mask occasional noises produced external to the testing room. An opaque curtain of black cloth divided the room to visually occlude the subject from the other corner. A digital recorder (ZOOM H4n Handy Recorder) was placed directly in front of the transfer cage at a distance of .76m. Audio signals were acquired at a sampling frequency of 96 kHz.

Every session typically consisted of two consecutive undirected experiments (one twin followed by the other) and one directed experiment (just one of the twins on a given day). Each session started with the undirected experiments lasting 5 min each. The order of the infants was counterbalanced. As soon as the undirected experiment was finished, one of the parents was brought to the experiment room and put into the opposing corner of the room. A second digital recorder (ZOOM H4n Handy Recorder) was placed directly in front of the parent at a distance of 0.76m from the transfer cage. During this setup procedure and throughout the directed experiment, the opaque curtain prevented the infant and the parent from having visual contact. The directed experiment lasted for $\approx 15$ min. The order of which parent participated in the interaction was counterbalanced. If the parent took more than 15 min to be lured for the directed calls experiment, the experiment was aborted to avoid any excessive separation stress on infants and parents. The number of undirected experiments with at least one call production was 40, 38, 38, 38, 37, 39, 19, 15, 16, 21 (10 infants, 301 sessions, 73,421 utterances). The number of directed experiments for each infant was 17, 13, 13, 18, 24, 24, 22, 21, 21, 22 (10 infants, 195 sessions). The number of subjects used in this study is based on a previous cross-sectional developmental study of marmoset vocalization that studied nine marmosets (*Pistorio et al., 2006*). A post hoc power analysis using G*Power 3.1 showed an achieved power of 0.818 for the correlation in *Figure 5f* ($n = 10$, Pearson's $r = -0.771$, Type I error $= 0.05$, $H_0 : r = 0$). All the experimental data used in this article is documented and can be found at http://science.sciencemag.org/content/suppl/2015/08/13/349.6249.734.DC1.

## Detection of calls

To determine the onset and offset of a syllable, a custom made MATLAB routine automatically detected the onset and offset of any signal that differed from background noise over a specific frequency range. To detect the differences, the full recording signal was first bandpass filtered between 6 and 10 kHz. Second, the signal was resampled to a 1 kHz sampling rate, a Hilbert transform was applied and its absolute value was calculated to obtain the amplitude envelope of the signal. The amplitude envelope was further low pass filtered to 50 Hz. A segment of the recording without any call (silent) was chosen as a comparison baseline. The 99th percentile of the amplitude value in the silent period was used as the detection threshold. Sounds with an amplitude envelope higher than the threshold were considered as possible vocalizations. Finally, to ensure that sounds other than call syllables were excluded, a researcher verified whether each detected sound was a vocalization or not, based on the spectrogram.

## Quantification of the dominant frequency

After detecting the onset and offset of calls, a custom made MATLAB routine calculated the dominant frequency of each syllable. The dominant frequency of a syllable was calculated as the average frequency at which the spectrogram had maximum power. A cubic spline curve was fitted to the population data using the MATLAB csaps function.

## Classification of type of call syllables

Each automatically detected call was manually classified as phee, phee-cry, subharmonic-phee, cry, twitter, and trill, based on the spectro-temporal profile measured by the spectrogram. To ensure validity of our classification procedure, 10 sessions chosen at random were classified by two different individuals and compared. The classification matched in more than 99.9% of the call syllables. The six call types show very distinct spectro-temporal profiles and can be easily classified by eye (*Pistorio et al., 2006*; *Bezerra and Souto, 2008*).

## Calculation of phee/cry ratio and zero-crossing day

For the directed calls experiments, a whole (i.e., multisyllabic) call was defined as any uninterrupted sequence of utterances of the same type (phee or cry) with previous offset to next onset separated by less than 500 ms (*DiMattina and Wang, 2006*; *Takahashi et al., 2013*). To quantify the developmental transition from cry to phee, for each session and subject, the ratio between the number of phees minus cries and the number of phees plus cries was calculated, *i.e.*,

$$\text{phee/cry ratio} = \frac{(\text{\# of infant phee calls produced} - \text{\# of infant cry calls produced})}{(\text{\# of infant phee calls produced} + \text{\# of infant cry calls produced})}. \tag{12}$$

A cubic spline curve was fitted to the phee/cry ratio data to obtain the phee/cry ratio curve. The zero-crossing day was defined as the first point at which the phee/cry ratio curve crossed zero, transitioning from a negative to a positive value. The zero-crossing day quantifies how quickly each infant transitioned from the cry-abundant initial period to phee-dominated later period.

## Contingent/non-contingent responses vs. zero crossing day

A parental call was classified as a contingent response to an infant call if the parental call onset was separated by less than 5 s from the infant call offset with no other call between them (*Takahashi et al., 2015*). To test if the contingent parental responses were related to how fast infants transition from cries to phees, we calculated the Pearson's correlation and the linear regression between the proportion of infant phees to which the parents responded before the zero-crossing day (total number of contingent parental responses before the zero-crossing day divided by the total number of infant phees in the period) and the zero-crossing day. To calculate the correlation, only the proportion of contingent parental responses that occurred before the zero-crossing day were included to be consistent with the causal ordering in which the possible cause (contingent parental response) happens before the effect (zero-crossing day). We used MATLAB csaps function to calculate the correlation and significance test.

## Biomechanical model of vocal apparatus

To investigate how nonlinearities in infant marmoset calls arise, and why they decline throughout development, we extended previous biomechanical models of the human speech production system. The resulting biomechanical model of the larynx and upper vocal tract is based on the one-mass model of Titze (*Titze, 1988*), which is simpler than earlier two-mass models (*Ishizaka and Flanagan, 1972*; *Herzel, 1993*; *Lucero, 1993*) and can produce a wide range of birdsong (*Amador and Mindlin, 2008*; *Perl et al., 2011*; *Amador et al., 2013*). In the next two sections we describe the model; further technical details are provided in the Appendix.

## The vocal fold model

*Titze (1988)* approximates vocal fold dynamics using two modes of vibration: lateral displacement of the tissues in the form of a mucosal wave, and a flapping motion due to out-of-phase oscillations at the entry and exit of the glottis (*Perl et al., 2011*). Titze's model uses the body-cover hypothesis, which proposes that laryngeal vibrations are governed by muscles and cartilage that determine its

geometry, and by its covering of soft tissue that allows waves to propagate in the direction of air flow. Bilateral symmetry in vocal fold oscillations is assumed, simplifying the system to a single degree of freedom oscillator of the form

$$m\ddot{x}(t) + b(x(t), \dot{x}(t))\dot{x}(t) + k(x(t), t)x(t) = f(x(t), \dot{x}(t), t),$$ (13)

where $m$ is the mass of the vocal folds and $x$, $\dot{x}$ and $\ddot{x}$ respectively their lateral displacement, velocity and acceleration; $b(x, \dot{x})\dot{x}$ and $k(x, t)x$ are nonlinear damping and stiffness forces, $f(x, \dot{x}, t)$ is the driving force due to lung air pressure, and $t$ denotes time.

As we shall see, the functions $b(x, \dot{x})$ and $k(x)$ determine the kinds of dynamics produced, and they are typically written as power series. Even truncating these series at third order leaves many coefficients to be determined, and we therefore make a nonlinear change of coordinates that transforms *Equation (13)* to its *normal form* that appears in *Figure 3a*:

$$\dot{x} = y,$$ (14a)
$$\dot{y} = -\alpha(t)\gamma^2 - \beta(t)\gamma^2 x + \gamma^2 x^2 - \gamma xy - \gamma^2 x^3 - \gamma x^2 y.$$ (14b)

Here the number of coefficients or control parameters is reduced to 3. Normal forms preserve all qualitative aspects of the dynamics of the original system in the neighborhoods of critical parameter values where *bifurcations* (*Guckenheimer and Holmes, 1983*) occur and different dynamical behaviors appear. That this could be done for *Equation (13)* was first realized by *Perl et al. (2011)*. In this case the parameters $\alpha(t)$ and $\beta(t)$ (which may vary with time) represent lung air pressure and vocal fold tension, and $\gamma$ is a time constant. Details on the derivations of *Equations (13) and (14)* are provided in the Appendix.

Models such as *Equations (13) and (14)* have been fitted to experimental data and model simulations have been compared with human vocalization and bird song (*Mergell et al., 2000*; *Sitt et al., 2008*; *Zañartu et al., 2011*; *Amador et al., 2013*). However, vocal production in marmosets has not been extensively studied and detailed measurements of lung pressure and muscle activity are lacking. As a proxy for this data, recordings of different marmoset calls were used to fit model parameters in the present work. The relative simplicity of the normal form (14) is helpful in this regard.

## From vocal fold vibrations to calls

Equipped with a simple model of laryngeal dynamics, we next derive the resulting sound pressure signals emitted from the mouth. Again seeking simplicity, we appeal to source-filter theory, which assumes that the vocal fold dynamics are independent of filtering within the upper vocal tract (*Titze, 1994*). The derivation of Titze (*Titze, 1988*), outlined in Appendix §§1.1.1-1.1.2, shows that the pressure $P_{glottal}$ at entry to the upper vocal tract is proportional to $x(t)$ at the midpoint vocal fold position. *Figure 9a* shows phase space plots of $x$ and $y$ for air pressure and vocal fold tension corresponding to a cry (left) and a phee (right). *Figure 9b* shows the corresponding time histories of $x(t)$. The resulting pressure changes propagate through the upper vocal tract and mouth cavity, which we model as a uniform cylinder. At the exit from the cylinder, part of the wave is reflected back towards the entrance (glottis) and the rest is transmitted as sound. Letting $T/2$ be the time for sound to travel the length, $L$, of the cylinder, the supraglottal pressure, $P_{in}$, at the inlet to the upper vocal tract has the following form:

$$P_{in}(t) = f(x(t)) - rP_{in}(t - T),$$ (15)

where $f(x(t))$ is a function of $x(t)$ (Appendix §1.1.2) and $r \in [0, 1]$ is the reflection coefficient. Near any given point $x(t)$ the time-dependent function $f(x(t))$ may be approximated by a Taylor series, and ignoring second and higher order terms we obtain

$$P_{in}(t) = cx(t) - rP_{in}(t - T), \text{ where } cx(t) = P_{glottal},$$ (16)

and $c$ is a nonnegative constant. In *Takahashi et al. (2015)* a third order approximation was used (see Appendix 1.1.2, *Equation (34)*), but given that the higher order terms are small and produce only minor effects, here we use only the first order term.

Note that the vocal fold dynamics $x(t)$ determined by **Equation (14)** are independent of sound pressure in the vocal tract, but the incoming pressure $P_{in}(t)$ is affected by the reflection $rP_{in}(t-T)$. Finally, the emitted sound is the part not reflected back towards the vocal folds:

$$P_{sound}(t) = (1-r)P_{in}(t-T/2) \tag{17}$$

**Figure 9c** shows the signals $P_{sound}$ which result from the effect of resonance on $P_{glottal}$ in **Figure 9b** for comparison with waveforms from examples of a cry and phee recorded from an infant marmoset in **Figure 9d**.

Unlike the zebra finch song model of Amador et al. (**Amador et al., 2013**), we do not model the mouth cavity separately; our model can reproduce typical marmoset calls well without this refinement, as shown in **Figure 3b–e**. Thus, the mathematical model is defined by **Equations (14) and (16–17)**. The components of the model are summarized in **Figure 3a** in relation to those of the marmoset's vocal apparatus in panel (a), and **Table 1** lists parameter values and ranges used in simulations. To verify that the model could reproduce realistic marmoset calls, the parameters were fit manually to match the spectrotemporal data of calls as shown in **Figure 3b–e**.

## Numerical simulations

Numerical simulations of **Equations (14) and (16–17)** were carried out using Euler's method in custom written MATLAB codes. Parameter values are given in **Table 1**. To generate the simulated calls, we varied $\alpha(t)$ and $\beta(t)$ within the range $[0, 1.1]$ and matched the frequency spectra and temporal profiles of the simulated sound to the corresponding vocalizations. To improve the fit between the model and recordings, pink noise was added to the simulation to match its presence in the background of the exemplar vocalizations in **Figure 3b–e**, using the MATLAB pinknoise function (file exchange #42919 by Hristo Zhivomirov [**Kasdin, 1995**]). The parameter $\beta$ was held fixed for the cry, while $\alpha(t)$ was ramped up and down in a piecewise-linear manner; for the other calls, both $\alpha(t)$ and $\beta(t)$ were ramped up and down to produce the varying fundamental and harmonic frequencies of calls such as those in **Figure 3c–e**. High pass filtering of $P_{sound}(t)$ was done using MATLAB eegfilt.

Below, we provide the MATLAB code used to solve **Equations (14) and (16)**.

```
function [x, y, p_in] = funcamador(gamma, a, b, r, T, c, x1, y1, dt)
%FUNCAMADOR.M This function will use the Euler method to simulate the
%motion of the vocal folds.
%    This simulation will run for 1 s with time step dt and with
%    initial conditions x1 and y1.
%% Initializing system
t = 1000;
N = floor(t/dt);
x = zeros(1,N + 1);
y = zeros(1,N + 1);
x(1) = x1;
y(1) = y1;
p_in = zeros(1,N + 1);
%% Simulating using Euler
for n = 1:N
   x(n + 1) = x(n) + dt*y(n);
   y(n + 1) = y(n) + dt*(-a*gamma^2 - b*gamma^2*x(n) + gamma^2*x(n)^2 ...
    - gamma*x(n)*y(n) - gamma^2*x(n)^3 - gamma*x(n)^2*y(n) );
  if n < T + 1
     p_in(n + 1) = c*x(n);
  else
     p_in(n + 1) = c*x(n) - r*p_in(n-T);
  end
end
end
```

## Dynamics of the biomechanical model

Combination calls like that of *Figure 3e* suggest that infants can dynamically modulate their vocal output by relatively small muscular changes, since switches between the call types occur very rapidly (*Zhang and Ghazanfar, 2016*). We show that small changes in air pressure ($\alpha$) and laryngeal muscle tension ($\beta$) can switch our model's output from cries to phees.

*Figure 10* illustrates the vocal fold dynamics produced by *Equation (14)* over a range of values of air pressure $\alpha$ and muscle tension $\beta$. The top left panel shows curves in $(\alpha, \beta)$-space on which bifurcations of fixed points (equilibria) of this equation occur. As parameter values $(\alpha, \beta)$ cross these curves, equilibria appear or disappear, their stability types change, and limit cycles representing sustained periodic oscillations in $x(t)$ can appear, as illustrated in the phase portraits corresponding to regions I-V. Appendix §1.1.4 details the calculations that yield the bifurcation curves.

Only in the shaded region I, lying above the upper saddle-node bifurcation curve and to the right of the Hopf bifurcation curve, do robust stable limit cycles and hence calls exist, to which almost all solutions converge. Each passage around the cycle corresponds to the vocal folds opening and closing once. Moreover, as $(\alpha, \beta)$ approach the saddle-node bifurcation curve from above, the period of oscillations grows to infinity, so that small changes in these parameters can produce large changes in waveform and hence spectral content. In this region $x(t)$ varies rapidly and slowly in different parts of the cycle, implying a broad frequency content (see *Figure 9a,b* (left) above). This extreme sensitivity is responsible for the rapid switches from cries to phees as lung pressure and/or vocal tension increases.

No other regions reliably yield calls. Values of $\alpha < 0$ cannot produce sustained oscillations, because in region V there is a single stable equilibrium and in region IV there two stable equilibria and a saddle point; in both cases all solutions converge on equilibria and no sound is produced, consistent with the biological intuition that low driving pressure produces no sound. In region II one stable and one unstable equilibrium coexist with a saddle, again without limit cycles, and all solutions approach equilibria. A small stable limit cycle surrounds the unstable equilibrium in region III, but random perturbations due to noise typically drive the system to the stable equilibrium, thus quenching the oscillations.

We therefore focus on parameter values in region I. Since *Equation. (14)* only captures the behavior of the original vocal fold model of *Titze (1988)* locally (*Figure 11*; see Materials and methods: The vocal fold model), we restrict the control parameters to $0 < \alpha < 1.1$ and $0 < \beta < 1.1$ and use numerical simulations to fit values of $\alpha$, $\beta$ that produce the spectrograms and waveforms of calls of interest. The remaining parameters $\gamma$, $c$, $r$ and $T$ were chosen to reproduce observed resonant frequencies and sound pressure levels, as described in Appendix §1.1.2, and were fixed at the values listed in *Table 1*, unless otherwise specified.

## Parameter dependence of the calls

Within the parameter ranges that stably produce calls, we investigated the relationship between parameter values and characteristics of the resulting model call. To obtain these, we iterated over many parameter values, recorded the natural frequency and amplitude of the calls produced, and computed their gains $g(\alpha, \beta)$ and $g(\theta)$, as shown in *Figure 3i*.

## Fitting the resonance frequency and estimating the vocal tract length

For a closed-closed tube, the fundamental frequency is given by $f_0 = \frac{c_{sound}}{2L} = \frac{1}{T}$ Hz (*Kinsler and Frey, 1962*), which provides the relationship $L = c_{sound} T / 2$. We used $c_{sound} = 350$ m/s. We then calculated the resonance frequencies of the biomechanical model for upper vocal tract lengths $L = 7.9, 8.7, 9.6, 10$ mm and interpolated the frequency over this range with a cubic spline curve, thus relating $L \in [7.9, 10]$ to the resonance frequencies $1/T$. Using a second cubic spline curve fitted to the marmoset data and the relationship between $L$ and the resonance frequencies obtained previously, we calculated the corresponding vocal tract lengths $L$. For the data we used the dominant (highest amplitude) frequencies as surrogates of resonance frequency. The 95% confidence interval for the resulting estimated vocal tract lengths were calculated from the dominant frequency data by resampling with replacement 1000 times and repeating the estimation method on each of the resampled data sets.

**Table 1.** Parameter values used for simulations to fit marmoset calls. The notation [0, 1.1] means that values are chosen in the range 0 to 1.1.

| Parameter | Description | Value(s) |
|---|---|---|
| $dt$ | Time step size ($\mu$s) | 5 |
| $\alpha$ | Nondimensional pressure | [0, 1.1] |
| $\beta$ | Nondimensional muscle tension | [0, 1.1] |
| $\gamma$ | Time constant (1/ms) | 45 |
| $c$ | Pressure coefficient | 1 |
| $r$ | Pressure reflection coefficient | 0.8 |
| $T/2$ | Time for one way sound travel in vocal tract ($\mu$s) | 50 |

## Classifying the type of call in the model

Initially, the calls produced by the model were classified manually as had been done with the infant recordings (*Takahashi et al., 2015*). To facilitate analysis of the model, an automatic classifier was developed and manually validated on a smaller sample. For each pair ($\alpha$, $\beta$), the call was simulated for one second. The envelope was calculated using the Hilbert transform of the call lowpass filtered at 4 kHz. Then, the power spectrum of the modeled call and that of its envelope were calculated. Cries are generated by a combination of slow vocal fold vibrations and resonance (*Figure 9* (left)). Therefore, we expect the power spectrum of the amplitude modulation to contain a peak for the cry but not for the subharmonics or phee. To differentiate between the latter we compared the relative power of the first and the second peak of the power spectrum of the call itself. For the phee, we expect the first peak to be $f_0$, and hence the largest peak in the spectrum. In contrast, for subharmonics we expect peaks of the power spectrum to occur below the resonance frequency, so the largest peak will not be the first. As a result, if the first peak was larger than the second peak, then the call was classified as a phee and otherwise as a subharmonic.

## Softmax action selection rule

The softmax action selection rule is obtained by applying the maximum entropy principle. We state the principle in a simplified form that suffices for this article. Let $C : \Theta \to \mathbf{R}$ be a cost function which may also be called an 'observable' or 'utility' of the system. Assume that $C$ has expected value $\int_\Theta C(\theta)p(\theta)\mathrm{d}\theta = E$. Given a cost function $C$, a natural question is to know what is the probability distribution $p(\theta)$ that the animal will execute a specific action. In our case, knowing the cost $C$ of producing a vocalization, we ask what is the probability that a marmoset will produce a call with air pressure and laryngeal tension $\theta$. The maximum entropy principle specifies that the probability density $p$ associated with the cost function $C$ should be the one that maximizes the entropy $H(p) = -\int_\Theta p(\theta)\log p(\theta)\mathrm{d}\theta$ and satisfies the expectation constraint $\int_\Theta C(\theta)p(\theta)\mathrm{d}\theta = E$. In other words, the maximum entropy principle states that given a cost function and a constraint, we must choose a probability distribution that makes the fewest possible assumptions (because maximal ignorance equates to maximal entropy). Such a probability distribution is said to follow the softmax action selection rule and can be written as $p(\theta) = \exp(-\eta C(\theta))/Z$, where $Z$ is a normalizing constant so that $\int_\Theta p(\theta)\mathrm{d}\theta = 1$ and $\eta$ is chosen to satisfy the constraint on the expected value of $C$. Probabilities were computed using a right Riemann sum approximation.

A complementary way to understand the softmax action selection rule is to introduce gradient dynamics with a potential derived from the cost function $C$ (*Video 1*). More precisely, consider the diffusion equation

$$\mathrm{d}\theta_t = \frac{\partial C}{\partial \theta}(\theta_t)\mathrm{d}t + \sqrt{2/\eta}\,\mathrm{d}W_t, \tag{18}$$

where $\partial C/\partial\theta$ is the gradient of $C$ and $W_t$ is a Wiener process. The equilibrium probability distribution for the dynamics $\theta_t$ of *Equation (18)* is given by *Equation (3)*:

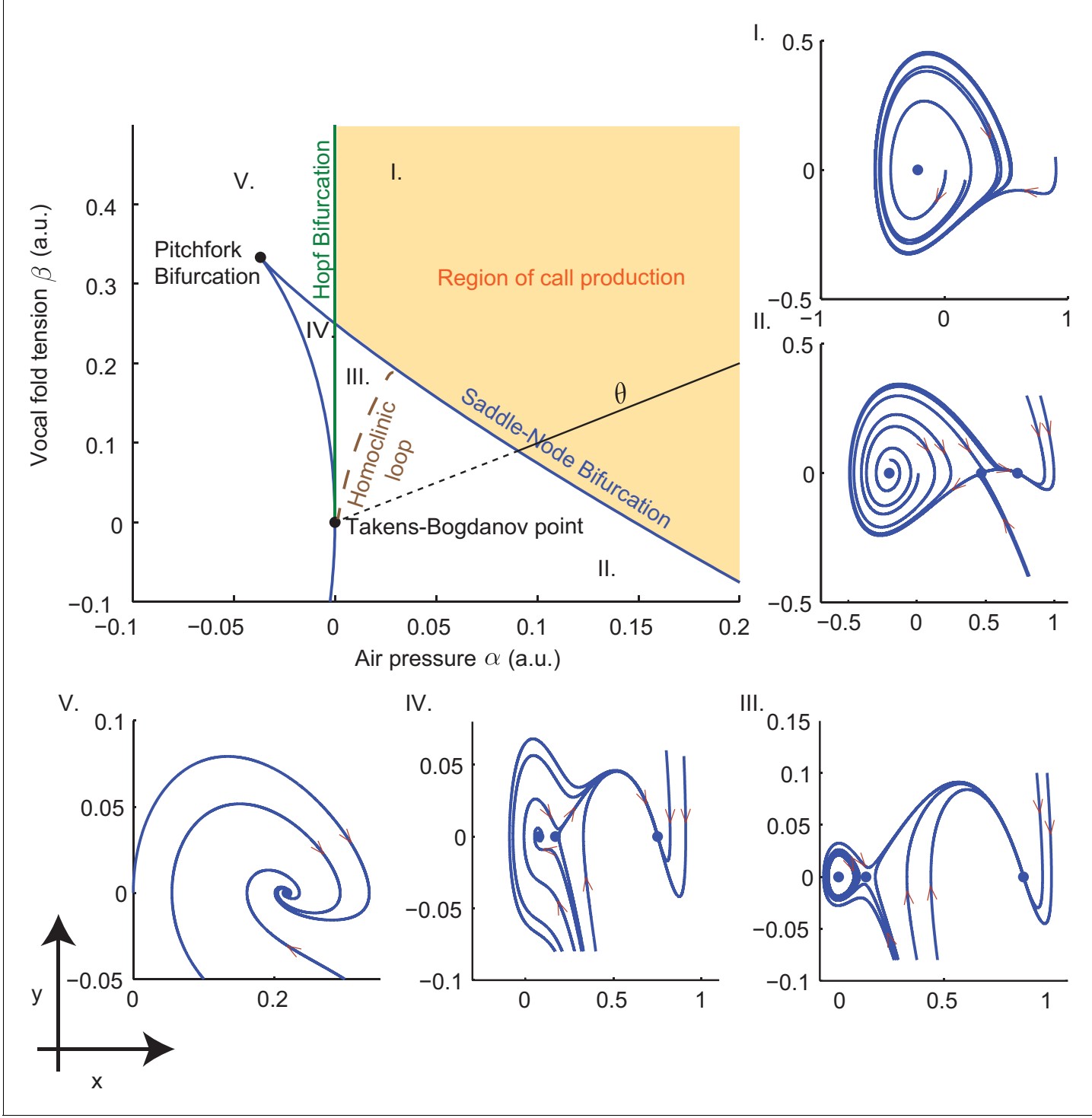

**Figure 10.** Bifurcation set and phase portraits of the model (*Equation (14)*). Top left panel shows the bifurcation set in the parameter space spanned by air pressure and muscle tension $(\alpha, \beta)$. Solid curves indicate saddle-node bifurcations in which pairs of fixed points disappear leaving regions II, III and IV, and Hopf bifurcations in which a stable limit cycle appears entering region I from region V and region III from region IV. Phase portraits in $(x, y)$-space illustrate vocal fold dynamics in regions I-V. Sustained oscillations surrounding a source produce calls in region I; a source, sink and saddle coexist with a small limit cycle in region III, but viable calls are not produced. A unique sink exists in region V, two sinks and a saddle in region IV, and a sink, saddle and source in region II; no sustained oscillations appear in these regions. Solid part of the line labeled $\theta$ starting at the Takens-Bogdanov point indicates the axis used in evaluating cost functions. Note that region of $(\alpha, \beta)$-parameter space is smaller than that in *Figure 3f–i*.

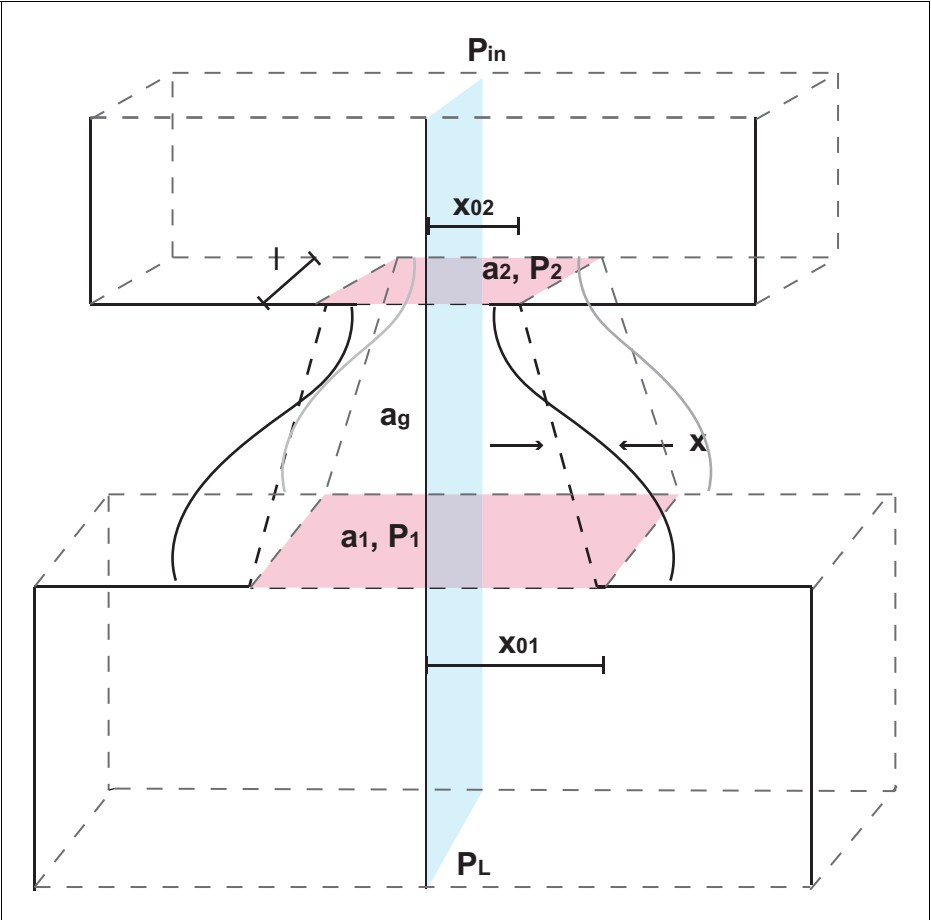

**Figure 11.** The larynx and glottis model. The coordinate system is shown with fixed depth $l$, lateral displacement $x(t)$ at midpoint, cross sectional areas $a_1, a_2$ at larynx entry and exit, $a_g$ at midpoint, air pressures $P_1, P_2$, and prephonatory widths $x_{01}, x_{02}$ at entry and exit. Adapted from *Titze (1988)*.

$$p(\theta) = \exp(-\eta C(\theta))/Z. \tag{19}$$

The resulting diffusion process therefore has an equilibrium measure given by the probability distribution predicted by the softmax action selection rule. If vocalizations are produced at periodic intervals (approximately once per second in infant marmosets [*Zhang and Ghazanfar, 2016*]), with the parameter defined by the stochastic process $\theta_t$, then the probability of producing each call type is given by the time that $\theta_t$ spends in each valley of the cost function $C_t(\theta)$. This probability is found by integrating $p(\theta)$ over the parameter region defining each call type, as in *Equations (5) and (6)*.

To generate the simulations for *Video 1*, we approximated the diffusion process (18) by a random walk with a potential that is a discretization of $C$. To allow rapid visualization of the typical diffusion dynamics, we arbitrarily sped up the timescale.

## Calculating $\lambda$ from the data in *Figure 5*

The cost function that only includes the contribution of vocal apparatus and muscle control is given by

$$C(\theta) = -\log g(\theta) + \lambda\theta. \tag{20}$$

To estimate the values of $\lambda$ for each postnatal day, we first fitted a cubic spline curve to the marmosets' phee/cry ratio data. Then we calculated the value of $\lambda$ that best approximated the phee/cry ratio curve for each postnatal day. The exact values of $\lambda$ depend on the choice of $\eta$, but the

**Table 2.** Parameter values used to plot the developmental landscapes in *Figure 8*.

| Parameter | Figure 8a | Figure 8b | Figure 8c | Figure 8d |
|---|---|---|---|---|
| $T/2$ $(\mu s)$ | 50 | 40, 45, 50 | 50 | 50 |
| $\eta$ | 300 | 300 | 300 | 300 |
| $\lambda_0$ | 3 | 3 | 3 | 3 |
| $\kappa$ $(\mathrm{days}^{-1})$ | 0.2126 | 0.2126 | 0.2126 | 0.2126 |
| $\delta$ $(\mathrm{days}^{-1})$ | 0.0654 | 0 | 0.0250, 0.0333, 0.0417 | 0.0417 |
| $F$ | 0.122 | 0 | 0 | 0.1176, 0.1566, 0.1961 |

difference is only in the scaling factor and the result in *Figure 5e* is representative for any choice of $\eta$. In *Figure 5c,e* we used $\eta = 5$. Larger values of $\eta$ have a similar effect, but since the probability densities are more concentrated on the peaks it is harder to display the effects of different $\lambda$'s in analogues of *Figure 5c,e*.

## The full cost function and more parameter choices

The final time-varying cost function with all its parameters can be written as

$$C_t(\theta) = -\log g_t(\theta) + \lambda_0 \theta - \delta t\theta - \kappa F t\theta. \tag{21}$$

We can decompose $C_t(\theta)$ as follows. The biomechanical contribution is represented by $g_t(\theta)$, where the dependency on time $t$ comes from the change in the vocal tract length $L$ or equivalently from the time $T/2 = L/c_{sound}$ for sound to traverse the cylinder. The change in muscle development is represented by $\delta$, the change in the nervous system by $\kappa$, and $F$ represents the contribution of social feedback. To obtain the sharp transition from low to high phee/cry ratio, a good value was $\eta = 300$ (*Figure 6b*). With this parameter, the zero-crossing day ($z_0$) occurred when $\lambda_z = 0.7927$, implying that

$$\frac{\lambda_0 - \lambda_z}{z_0} = \kappa F + \delta. \tag{22}$$

We chose $\lambda_0 = 3$ so *Equation (22)* yields $z_0 = \frac{2.2073}{\kappa F + \delta}$ (see *Figure 6c*). Any value of $\lambda_0 > \lambda_z$ would give the same curve fitting as we need only to rescale $\kappa$ and $\delta$ accordingly. $\lambda_0$ is the only parameter that cannot be estimated from the data. Fitting the function to the data relating the amount of parental feedback ($F$) and zero-crossing day $z_0$, we get $\kappa = 0.2126$ and $\delta = 0.0654$. *Table 2* lists the parameters used to produce *Figure 8b–e*.

## Correlating $F$ with $W$ and $N$

We tested if the rate of weight change $W$ and the rate of infant phee production $N$ before the zero-crossing day could predict the frequency of parental feedback $F$. To calculate the weight change we first calculated the difference between two consecutive weight measurements and divided by the number of days between them to obtain the local rate of weight change. The overall rate of weight change was calculated as the average of local rates of weight change before the zero-crossing day. If there were a linear relationship between the weight change and the frequency of parental feedback, we would expect a significant Pearson correlation ($r$) between these parameters. We also fitted a multiple linear regression between the explanatory variables $W$ and $N$, and the dependent variable $F$. We applied the two-sided t-test to verify the nullity of regression coefficients ($n = 10$ infants). We concluded that neither infant weight increases nor changes in phee call numbers could predict the frequency of parental feedback.

## Acknowledgements

We thank Morgan Gustison, Talmo Pereira and Nicholas Roy for their constructive comments on earlier drafts of this paper. This work was supported by NSF-CRCNS DMS-1430077 (PH), a James S McDonnell Scholar Award (AAG), and NIH R01NS054898 (AAG).

## Additional information

### Funding

| Funder | Grant reference number | Author |
|---|---|---|
| National Science Foundation | DMS-1430077 | Philip Holmes |
| National Institutes of Health | R01NS054898 | Asif A Ghazanfar |
| James S. McDonnell Foundation | 220020238 | Asif A Ghazanfar |

The funders had no role in study design, data collection and interpretation, or the decision to submit the work for publication.

### Author contributions

YT, Conceptualization, Formal analysis, Methodology, Writing—original draft, Writing—review and editing; DYT, Conceptualization, Data curation, Formal analysis, Supervision, Investigation, Visualization, Writing—original draft, Writing—review and editing; PH, Conceptualization, Formal analysis, Supervision, Funding acquisition, Investigation, Methodology, Writing—original draft, Writing—review and editing; AAG, Conceptualization, Supervision, Funding acquisition, Validation, Investigation, Writing—original draft, Writing—review and editing

### Author ORCIDs

Yayoi Teramoto, http://orcid.org/0000-0003-3419-0351
Asif A Ghazanfar, http://orcid.org/0000-0003-1960-7470

### Ethics

Animal experimentation: This study was performed in strict accordance with the recommendations in the Guide for the Care and Use of Laboratory Animals of the National Institutes of Health. All of the animals were handled according to approved institutional animal care and use committee (IACUC) protocols (#1908-15) of Princeton University.

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

# Appendix 1 Derivation of the biomechanical model

In this appendix we review background material on laryngeal dynamics and vocal production models, and provide details on the derivation of the biomechanical model and the normal form of the vocal fold equations used in this paper.

## 1.1 The biomechanical model

As in recent work on songbirds (*Perl et al., 2011*; *Amador et al., 2013*), we base our model on Titze's work on human vocal fold dynamics and voice production (*Titze, 1988*). We briefy summarize conservation laws from fluid mechanics (e.g. *Bird et al., 2007*, Chap. 3), which are used in deriving the model, and then outline its simplification via coordinate transformations to obtain the normal form describing vocal fold dynamics (*Equation (14)* in the main text):

$$\dot{x} = y, \tag{23a}$$
$$\dot{y} = -\alpha\gamma^2 - \beta\gamma^2 x + \gamma^2 x^2 - \gamma xy - \gamma^2 x^3 - \gamma x^2 y. \tag{23b}$$

In the process we find that the usual procedure, using Taylor series expansion about a degenerate (codimension 2) fixed point with a double zero eigenvalue and assuming linear and cubic order stiffness and damping terms, yields the term $+2\gamma xy/3$ instead of $-\gamma xy$ as in *Perl et al. (2011)*, *Amador et al. (2013)*, and we indicate a modification to the damping term in the original model of *Titze (1988)* that would produce *Equation (23)*. We explain the reason for this difference and also derive explicit expressions in terms of the parameters $(\alpha, \beta)$ for which bifurcations create steady and periodic states relevant for vocalizations.

### 1.1.1 Conservation laws for one-dimensional flows

Bernouilli's law expresses the conservation of energy in a fluid flowing at velocity $v$ through a frictionless pipe:

$$\frac{P}{\rho} + gz + \frac{v^2}{2} = \text{constant}, \tag{24}$$

where $P$ and $\rho$ denote fluid pressure and density, $g$ the acceleration due to gravity, and $z$ the height of the pipe. Variations of gravitational energy $gz$ in the respiratory and vocal system are insignificant, and we need only consider the balance of potential and kinetic energy in the first and third terms of (24). The mass flow rate $\dot{M} = \rho v a$ also remains constant but cross sectional areas $a$ can change. Across such a change from area $a_1$ to $a_2$, these conservation laws imply that

$$P_1 + \frac{1}{2}\rho v_1^2 = P_2 + \frac{1}{2}\rho v_2^2 \text{ and } a_1 v_1 = a_2 v_2, \tag{25}$$

where the subscripts denote values in the locations of areas $a_1$ and $a_2$. In writing *Equations (25)* we have also assumed that changes in density are negligible ($\rho = \text{constant}$). *Titze, 1988*, uses these equations to express pressure differences across the vocal folds.

## 1.1.2 Vocal fold dynamics and coupling to the upper vocal tract

We describe vocal fold dynamics in terms of two modes: displacement of tissue in the direction of airflow, and lateral flapping due to antiphase motions at the entry and exit of the larynx. The tissue displacement is approximated by a traveling mucosal surface wave of fixed shape, as in d'Alembert's solution of the classical wave equation (*Greenberg, 1978*), that couples time and space dependence. This implies that the lateral displacements $x_1(t)$ and $x_2(t)$ at entry and exit of the larynx can be written in terms of the displacement $x(t)$ and velocity $\dot{x}(t)$ at the midpoint of the vocal folds. Section IIA of *Titze (1988)* approximates the wave by a linear function, and letting $\tau$ denote the time taken for it to travel half the length of the glottis, obtains

$$x_1 = x + \tau\dot{x} \ \text{ and } \ x_2 = x - \tau\dot{x}. \tag{26}$$

The resulting cross-sectional areas $a_1, a_2$ at entry and exit of the glottis are therefore

$$a_1(t) = 2l(x_{01} + x_1) = 2l(x_{01} + x(t) + \tau\dot{x}(t)), \tag{27a}$$
$$a_2(t) = 2l(x_{02} + x_2) = 2l(x_{02} + x(t) - \tau\dot{x}(t)), \tag{27b}$$

where $l$ is the depth of the glottis and $x_{01}$ and $x_{02}$ are the prephonatory lateral positions of its lower and upper ends in the absence of airflow: see *Figure 11*. We do not model the flow in the trachea, assuming that it simply transmits lung pressure to the larynx (see below).

As indicated in *Figure 11* (cf. *Figure 3a*), we assume bilateral symmetry and represent the vocal folds, moving in antiphase on left and right, by a mass $m$ supported by a spring, subject to dissipative forces and driven by the transglottal pressure. The second order ordinary differential equation (ODE) that describes the resulting dynamics therefore has the form

$$m\ddot{x} + b(x, \dot{x})\dot{x} + k(x)x = a_g P_{av}(t), \tag{28}$$

where the nonlinear terms $b(x, \dot{x})$ and $k(x)$ represent energy dissipation and spring stiffness, and the last term is the driving force due to the mean pressure $P_{av}$ averaged over the glottis from entry to exit, applied to the glottal area $a_g$ measured at the midpoint of the vocal folds. Using *Equations (25)* and several simplifying assumptions, *(Titze, 1988)*, ( see § IIB-C, *Figure 5* and *Equation (21)* of the reference) shows that, when subglottal pressure $P_1$ is equal to lung air pressure $P_L$, vocal tract input pressure is atmospheric, and supraglottal area is large relative to $a_2$, $P_{av}$ can be approximated as

$$P_{av}(t) = \frac{P_L(t)}{k_t}\left(1 - \frac{a_2(t)}{a_1(t)}\right), \tag{29}$$

where $k_t$ is a kinetic pressure coefficient representing losses in the entry and glottis and recovery in the supra-glottal expansion region.

### Vocal fold dynamics

We follow *Perl et al. (2011)* in assuming third order nonlinearities in the stiffness and damping terms of *Equation (28)*, and using *Equation (29)* and *Equations (27a–27b)*, obtain the system

$$\dot{x} = y, \tag{30a}$$

$$\dot{y} = \frac{1}{m}\left[-(k_1 + k_2 x^2)x - (b + cx^2)y + \frac{a_g P_L(x_{01} - x_{02} + 2\tau y)}{k_t(x_{01} + x + \tau y)}\right]; \tag{30b}$$

cf. *Titze (1988)*, *Equation (22)* and *Perl et al. (2011)*, *Equations (1–3)*. Here the parameters $m, k_2, c, a_g, x_{01}, x_{02}, k_t$ and $\tau$ are positive and $k_1, b, P_L$ may take either sign. Note that some symbols differ from those of *Perl et al. (2011)* and that we have replaced the nonspecific damping term $b(y)y$ in *Perl et al. (2011)*, *Equation (1)* by the linear damping term $by$. *Equation (30)* contains 11 parameters, most of which are unknown, but our subsequent reduction to the normal form *Equation (23)* with an overall time scale and two nondimensional parameters analogous to $k_1$ and $P_L$ will allow us to fit fundamental call frequencies and explore the biophysical space of muscle tension and driving pressure. However, as shown below, to obtain this normal form, also given in *Perl et al. (2011)*, *Equation (8)*, may require inclusion of a quadratic damping term $xy$ or an additional forcing term (*Amador, 2009*). Henceforth we assume that all parameters are held constant, because the normal form theory used here applies only to autonomous ODEs.

## Coupling to the vocal tract

In *Titze (1988)*, *Equation (35)* models the fluctuating input pressure to the vocal tract as $P_{in} = R_2 u + I_2 \dot{u}$, where $u = a_2 v_2$ is the flow rate at glottal exit and $R_2, I_2$ are the tract's input resistance and inertance. The flow derivative is therefore $\dot{u} = \dot{a}_2 v_2 + a_2 \dot{v}_2$, but for simplicity we assume quasi-steady flow $\dot{v}_2 \approx 0$. Thus, from *Equation (27b)*, we obtain

$$\dot{u} = 2lv_2(\dot{x} - \tau\ddot{x}). \tag{31}$$

and we may write the input pressure in terms of the vocal fold displacement $x$ as

$$P_{in}(t) = 2v_2 l[I_2(\dot{x} - \tau\ddot{x}) + R_2(x_{02} + x - \tau\dot{x})]. \tag{32}$$

## Vocal tract dynamics

The supraglottal vocal tract and mouth cavity filter the input pressure $P_{in}(t)$. In the absence of details on marmosets, we model the entire supraglottal sytem as a cylinder supporting traveling plane waves, a fraction of which are reflected back from the exit at the mouth, and the remainder transmitted to produce the animal's calls. Letting $T = 2L/c_{sound}$ denote the time taken for waves to travel at speed $c_{sound}$ to the exit and back and $r \in [0, 1]$ be the reflection coefficient, this adds a delayed and scaled copy of the input pressure to *Equation (32)*:

$$P_{in}(t) = 2v_2 L[I_2(\dot{x} - \tau\ddot{x}) + R_2(x_{02} + x + \tau\dot{x})] - rP_{in}(t - T). \tag{33}$$

Collecting the terms multiplying $x, \dot{x}$ and $\ddot{x}$ yields the series approximation

$$P_{in}(t) = c_1 x(t) + c_2 \dot{x}(t) - c_3 \ddot{x}(t) - rP_{in}(t - T) \tag{34}$$

which appears as *Equation (16)* in the main text. For further details see *Titze and Alipour, (2006)*, Ch 7. The part of $P_{in}$ not reflected produces the transmitted sound pressure $P_{sound} = (1 - r)P_{in}(t - T/2)$: *Equation (17)* in the main text. We note that the parameters $c_1 = 1, c_2 = 0.01, c_3 = 0.001$ used by *Takahashi et al. (2015)* prompted our neglect of $c_2$ and $c_3$ in *Equation (16)* in the main text.

### 1.1.3 Normal form transformation

Here we explain how the simplified normal form ODE may be derived from the version of Titze's model of vocal fold dynamics adopted by *Perl et al. (2011)*, *Equations (30)* above. For a general introduction to normal forms and analyses of relevant examples, see Guckenheimer and Holmes (*Guckenheimer and Holmes, 1983*, §3.3 and §7.3).

In studying linear ODEs of the form $\dot{\mathbf{x}} = \mathbf{A}\mathbf{x}$, where $\mathbf{x}$ is an $n$-vector and $\mathbf{A}$ is an $n \times n$ constant matrix, it is helpful to change coordinates via a similarity transformation $\mathbf{x} = \mathbf{T}\mathbf{y}$, where $\mathbf{T}$ is a matrix whose columns are eigenvectors of $\mathbf{A}$. In the $\mathbf{y}$-coordinates, the ODE becomes $\dot{\mathbf{y}} = \mathbf{T}^{-1}\mathbf{A}\mathbf{T}\mathbf{y} \overset{\text{def}}{=} \Lambda y$ and $\Lambda$ is a diagonal matrix containing the eigenvalues of $\mathbf{A}$.

The transformation decouples the variables, making analyses much simpler; it also effectively reduces $n^2$ possible matrix entries to the $n$ eigenvalues. In a similar manner, normal form theory allows one to simplify nonlinear ODEs $\dot{\mathbf{x}} = \mathbf{f}(\mathbf{x})$, where the components of $\mathbf{f}(\mathbf{x})$ are polynomial functions.

We start by nondimensionalizing the displacement and velocity in *Equation (30)*, letting $x_1 = x/x_{01}, x_2 = y/x_{01}$ and renaming parameter groups for convenience. to obtain:

$$\dot{x}_1 = x_2, \tag{35a}$$

$$\dot{x}_2 = \gamma^2 \left[ -(\kappa + x_1^2)x_1 - (\delta_1 + \delta_2 x_1^2)\tau x_2 + \frac{\mathcal{P}(\epsilon + 2\tau x_2)}{(1 + x_1 + \tau x_2)} \right], \text{ where} \tag{35b}$$

$$\gamma = x_{01}\sqrt{\frac{k_2}{m}}, \kappa = \frac{k_1^2}{x_{01}^2 k_2}, \delta_1 = \frac{b}{x_{01}^2 k_2 \tau}, \delta_2 = \frac{c}{k_2 \tau}, \mathcal{P} = \frac{a_g P_L}{k_t x_{01}^3 k_2}, \text{ and } \epsilon = \frac{x_{01} - x_{02}}{x_{01}}. \tag{35c}$$

Here $\tau$ is the glottal timescale from *Equations (26)*, $\gamma$ is an overall inverse timescale, velocity $x_2$ also has dimension time$^{-1}$ and the remaining parameters are dimensionless. Note that all the parameters excepting $\kappa, \mathcal{P}$ and $\delta_1$ are strictly positive; in particular coefficients of the terms $x_1^3$ and $x_1^2 x_2$ on the right hand side of *Equation (35b)* must be negative for global stability of the vocal fold dynamics.

Fixed points of *Equations (35a–35b)* lie at $(x_1, x_2) = (\bar{x}_1, 0)$, where $\bar{x}_1$ solves the quartic equation

$$(1 + \bar{x}_1)(\kappa + \bar{x}_1^2)\bar{x}_1 = \mathcal{P}\epsilon, \tag{36}$$

and to obtain the normal form we will make a Taylor series expansion about a double root at which the Jacobian matrix of *Equations (35)* has a zero eigenvalue of multiplicity 2. For $\kappa<0$ and $\mathcal{P} = 0$ there are three biophysically relevant roots $\bar{x}_1 = 0$ and $\bar{x}_1 = \pm\sqrt{-\kappa}$ (the root $\bar{x}_1 = -1$ corresponds to a closed glottis). For small $\kappa = -\bar{\kappa}<0$ and

$$\mathcal{P} = \frac{2\bar{\kappa}^{\frac{3}{2}}}{3\sqrt{3}} - \frac{4\bar{\kappa}^2}{9} + \mathcal{O}\left(\bar{\kappa}^{5/2}\right) > 0, \tag{37}$$

a double root occurs at

$$\bar{x}_1 = -\sqrt{\frac{\bar{\kappa}}{3}} + \frac{\bar{\kappa}}{3} + \mathcal{O}\left(\bar{\kappa}^{\frac{3}{2}}\right). \tag{38}$$

If we additionally choose the dissipation parameters such that

$$\delta_1 + \delta_2 \bar{x}_1^2 = \frac{\mathcal{P}(2(1+\bar{x}_1)-\epsilon)}{(1+\bar{x}_1)^2}, \tag{39}$$

the Jacobian matrix of *Equation (35)* linearized at the degenerate fixed point is

$$\mathbf{A} = \begin{bmatrix} 0 & 1 \\ 0 & 0 \end{bmatrix}. \tag{40}$$

This linear part identifies a Takens-Bogdanov point (*Guckenheimer and Holmes, 1983*, §3.3,§7.3), see §1.1.4 below. Note that *Equation (39)* implies that $\delta_1 = \mathcal{O}(\bar{\kappa}^{3/2})$ and $\delta_2 = \mathcal{O}(\bar{\kappa}^{1/2})$. Thus, at this point the nondimensional pressure $\mathcal{P}$, and dissipation parameters $\delta_1, \delta_2$ are all small and scaled in fractional powers of the linear stiffness $\kappa$, which is also small and negative.

We now expand the right hand side of *Equation (35)* in a Taylor series about $(\bar{x}_1, 0)$ that may be written in vector notation as

$$\dot{\mathbf{x}} = \mathbf{A}\mathbf{x} + \mathbf{f}_2(\mathbf{x}) + \mathbf{f}_3(\mathbf{x}) + \mathcal{O}(|\mathbf{x}|^4), \tag{41}$$

where $\mathbf{x}$ denotes the deviation from $(\bar{x}_1, 0)$ and the $\mathbf{f}_j(\mathbf{x})$ are homogeneous polynomial functions of order $j$. In general, all possible terms $x_1^k x_2^{(j-k)}$ may occur in each of these functions: six in $\mathbf{f}_2$, eight in $\mathbf{f}_3$, etc. We next define a new variable $\mathbf{y}$ via the near-identity transformation

$$\mathbf{x} = \mathbf{y} + \mathbf{P}(\mathbf{y}) \underset{=}{\mathrm{def}} \mathbf{y} + \sum_{j=2}^{N} \mathbf{P}_j(\mathbf{y}), \tag{42}$$

where the functions $\mathbf{P}_j$ are homogeneous polynomials in $\mathbf{y}$. Under this change of variables *Equation (41)* becomes

$$\dot{\mathbf{y}} = \mathbf{A}\mathbf{y} + \mathbf{g}_2(\mathbf{y}) + \mathbf{g}_3(\mathbf{y}) + \mathcal{O}(|\mathbf{y}|^4), \tag{43}$$

in which the functions $\mathbf{g}_j$ can be significantly simplified by suitable choices of the $\mathbf{P}_j$'s.

Specifically, differentiating *Equation (42)* with respect to time and using the chain rule, we obtain $\dot{\mathbf{x}} = [\mathbf{I} + \mathbf{DP}(\mathbf{y})]\dot{\mathbf{y}}$ and thereby find that

$$\dot{\mathbf{y}} = [\mathbf{I} + \mathbf{DP}(\mathbf{y})]^{-1}[\mathbf{A}(\mathbf{y} + \mathbf{P}(\mathbf{y})) + \mathbf{f}_2(\mathbf{y} + \mathbf{P}(\mathbf{y})) + \mathbf{f}_3(\mathbf{y} + \mathbf{P}(\mathbf{y})) + \mathcal{O}(|\mathbf{y}|^4)], \tag{44}$$

where $\mathbf{DP}$ denotes the Jacobian matrix of first order derivatives of $\mathbf{P}$. As shown in *Equation (43)*, the first order (linear) term in this ODE is $\mathbf{A}\mathbf{y}$ (it is unchanged by the transformation $\mathbf{x} = \mathbf{y} + \mathbf{P}(\mathbf{y})$), but the quadratic order function is

$$\mathbf{g}_2(\mathbf{y}) = \mathbf{A}\mathbf{P}_2(\mathbf{y}) - \mathbf{DP}_2(\mathbf{y})\mathbf{A}\mathbf{y} + \mathbf{f}_2(\mathbf{y}). \tag{45}$$

We now seek to choose the six terms in the two components of the quadratic vector function $\mathbf{P}_2$ to cancel as many of the analogous terms in $\mathbf{f}_2$ as possible. The matrix $\mathbf{A}$ determines the extent to which this can be done via the *Lie Bracket* operator $\mathrm{adL}(\mathbf{P}_2) = [\mathbf{A}\mathbf{P}_2(\mathbf{y}) - \mathbf{DP}_2(\mathbf{y})\mathbf{A}\mathbf{y}]$, where $\mathbf{L} = \mathbf{A}\mathbf{y}$ denotes the linear part of *Equation (41)* and $\mathbf{DP}_j$ the Jacobian matrix of first partial derivatives of $\mathbf{P}_j$. In general $\mathrm{adL}(\mathbf{P}_j)$ is a homogeneous polynomial of order $j$, so transformations of increasing order can successively remove terms in the 'new' nonlinear functions $\mathbf{g}_j(\mathbf{y})$.

For the matrix (40) all quadratic terms except those of the forms

$$\begin{pmatrix} 0 \\ y_1^2 \end{pmatrix} \text{ and } \begin{pmatrix} 0 \\ y_1 y_2 \end{pmatrix} \tag{46}$$

can be removed, and a similar computation using $\mathbf{P}_3$ shows that all cubic terms except

$$\begin{pmatrix} 0 \\ y_1^3 \end{pmatrix} \text{ and } \begin{pmatrix} 0 \\ y_1^2 y_2 \end{pmatrix} \tag{47}$$

can be removed (*Guckenheimer and Holmes, 1983*, §3.3). These are precisely the four nonlinear terms that remain in the normal form *Equation (23)* adopted by *Amador et al. (2013)* (similar pairs of terms would remain at each higher order power). However, in performing the transformation $\mathbf{x} = \mathbf{y} + \mathbf{P}_2(\mathbf{y})$ to remove terms of $\mathcal{O}(|\mathbf{y}|^2)$ from $\mathbf{g}_2(\mathbf{y})$, additional terms of $\mathcal{O}(|\mathbf{y}|^3)$ and higher are introduced. Except for multiples of those in *Equation (47)*, these can be removed by a subsequent transformation $\mathbf{y} = \mathbf{z} + \mathbf{P}_3(\mathbf{z})$, but to obtain the correct ODE at cubic order terms of the forms (47) must be included. Since we will neglect all terms of $\mathcal{O}(|\mathbf{z}|^4)$, this analysis is strictly valid only for sufficiently small state variable values $(x_1, x_2)$ and in a neighborhood of the Takens-Bogdanov point, implying that the parameters $\kappa$ and $\mathcal{P}$ should also remain small.

We now describe the specific transformations employed to derive the normal form. As noted above we first change coordinates to shift the degenerate Takens-Bogdanov point $(\bar{x}_1, 0)$ occurring for the parameter values of *Equations (37–39)* to the origin. To avoid excessive notation, we retain the notation $x_1$ for the vocal fold displacement, now measured relative to $\bar{x}_1$:

$$\dot{x}_1 = x_2, \tag{48a}$$

$$\dot{x}_2 = \gamma^2 \left[ A x_1^2 + B x_1 x_2 + C x_2^2 + D x_1^3 + E x_1^2 x_2 + F x_1 x_2^2 + G x_2^3 + \mathcal{O}(|x_j|^4) \right], \tag{48b}$$

where

$$A = \frac{\mathcal{P}\epsilon}{(1+\bar{x}_1)^3} - 3\bar{x}_1, \; B = 2\left[ \frac{\mathcal{P}(\epsilon - (1+\bar{x}_1))}{(1+\bar{x}_1)^3} - \delta_2 \bar{x}_1 \right] \tau, \; C = \frac{\mathcal{P}(\epsilon - 2(1+\bar{x}_1))\tau^2}{(1+\bar{x}_1)^3},$$

$$D = -\left[ 1 + \frac{\mathcal{P}\epsilon}{(1+\bar{x}_1)^4} \right], \; E = -\left[ \delta_2 + \frac{\mathcal{P}(3\epsilon - 2(1+\bar{x}_1))}{(1+\bar{x}_1)^4} \right] \tau. \tag{49}$$

There are similar expressions for the coefficients $F$ and $G$ but since they can be removed from the $\mathcal{O}(|x_j|^3$ they are not needed here. At $\mathcal{O}(|x_j|^2$ the term $Cx_2^2$ can be removed by the transformation

$$\mathbf{x} = \mathbf{y} + \mathbf{P}_2(\mathbf{y}) \text{ with } \mathbf{P}_2(\mathbf{y}) = C \begin{pmatrix} y_1^2/2 \\ y_1 y_2 \end{pmatrix}, \tag{50}$$

which, via the Lie Bracket operation, produces the additional terms

$$\begin{pmatrix} 0 \\ BCy_1^2 y_2/2 \end{pmatrix} \text{ and } \begin{pmatrix} 0 \\ 2C^2 y_1 y_2^2 \end{pmatrix} \tag{51}$$

at $\mathcal{O}(|x_j|^3)$. The second of these can be removed along with the terms $Fy_1 y_2^2$ and $Gy_2^3$ by a further transformation $\mathbf{y} = \mathbf{z} + \mathbf{P}_3(\mathbf{z})$, but the first must be added to the term $Ey_1^2 y_2$ which

which passes unchanged through the transformation from **Equation (48b)**. This yields the ODE

$$\dot{y}_1 = y_2, \tag{52a}$$

$$\dot{y}_2 = \gamma^2 \left[ A y_1^2 + B y_1 y_2 + D y_1^3 + \left( E + \frac{BC}{2} \right) y_1^2 y_2 + \mathcal{O}(|y_j|^4) \right], \tag{52b}$$

Examining the parameter combinations that appear in the coefficients $A, \ldots, E$ via **Equations (49) and (35c)**, using the asymptotic expressions (37–38), continuing to assume that $\mathcal{P}$ and $\kappa = -\bar{\kappa}$ remain small, and retaining only the leading order $\mathcal{O}(\sqrt{\bar{\kappa}})$ and $\mathcal{O}(1)$ terms, we deduce that $A \approx \sqrt{3\bar{\kappa}} > 0$, $B \approx 2\delta_2 \tau \sqrt{\bar{\kappa}/3} > 0$, $D \approx -1 < 0$ and $E + BC/2 \approx -\delta_2 \tau < 0$, consistent with global stability as noted following **Equations (35)**. Setting $\bar{\kappa} = 1/3$ and $\delta_2 \tau = 1/\gamma$ and letting $-\alpha, -\beta$ denote departures in nondimensional pressure $\mathcal{P}$ and linear stiffness $\kappa$ (muscle tension) from the values corresponding to the Takens-Bogdanov point, we obtain the normal form (23) with the *exception that the coefficient $\gamma^2 B$ of $xy$ is $+2\gamma/3$*, in place of $-\gamma$.

The factor $2/3$ is not crucial, but the fact that the quadratic damping term has the opposite sign to that of **Equation (23)** removes a key feature of the model, namely, the occurence of a saddle-node bifurcation on a limit cycle on the upper saddle-node bifurcation curve shown in **Figure 10** and described in the next section. Without the long-period finite amplitude limit cycles that exist near this curve, the model does not produce infant cries with broad spectral content. This region in parameter space is also where all but one of the 'gestures' in zebra finch song are located (**Amador et al., 2013**, **Figure 2d**).

To obtain a negative sign for $B$, it suffices to add a quadratic damping term, replacing $(\delta_1 + \delta_2 x_1^2)\tau x_2$ in **Equation (35b)** by $(\delta_1 + \bar{\delta} x_1 + \delta_2 x_1^2)\tau x_2$, which modifies the coefficient to $\gamma^2 B = \gamma^2(\bar{\delta} + 2\delta_2\sqrt{\bar{\kappa}/3})\tau$. The left hand side of the dissipation balance (39) becomes $\delta_1 + \bar{\delta}\bar{x}_1 + \delta_2 \bar{x}_1^2$ and setting $\bar{\delta}\tau = 1/\gamma$ then yields the normal form (23) at leading order. We note that the term $-\bar{\delta}x_1 x_2$ implies that the force is negative when $x_1 > 0$ (during vocal fold opening), but positive when $x_1 < 0$ (vocal fold closing), leading to a balance of energy dissipation and creation. Similar effects arise from terms of the form $\bar{\delta}x_2^2$ (**Holmes, 1977**). Also, the linear damping term $\delta_1 x_2$ could be omitted without affecting the normal form.

We note that in her PhD thesis (**Amador, 2009**) includes a constant term in **Equation (30b)** representing the force due to dorsal gating muscles [G.B. Mindlin, personal communication]. Adjusting this additional parameter allows one to locate the Takens-Bogdanov point more easily and also derive the normal form with a positive coefficient in the $xy$ term, as used in **Perl et al. (2011)** and **Amador et al. (2013)**.

## 1.1.4 Bifurcation sets and parameter dependence

We now derive the bifurcation set of the normal form ODEs **Equation (23)** and infer the qualitative structures of the phase portraits shown in **Figure 10**. For this analysis we take the pressure and muscle tension parameters $(\alpha, \beta)$ as time-independent and seek steady solutions, specifically fixed points and limit cycles. For $\gamma > 0$, fixed points $(x, y) = (\bar{x}, 0)$ of **Equations (23)** occur with $\bar{x}$ satisfying the cubic equation

$$\bar{x}^3 - \bar{x}^2 + \beta\bar{x} + \alpha = 0; \tag{53}$$

one, two or three fixed points can exist, depending on the value of $(\alpha, \beta)$.

Stability types of the fixed points are determined by the eigenvalues of the Jacobian matrix

$$\mathbf{J} = \begin{bmatrix} 0 & 1 \\ \gamma^2(-\beta + 2x - 3x^2) - \gamma(2xy + y) & -\gamma x(x+1) \end{bmatrix}_{(\bar{x},0)} \tag{54a}$$

$$= \begin{bmatrix} 0 & 1 \\ -\gamma^2(\beta - 2\bar{x} + 3\bar{x}^2) & -\gamma\bar{x}(\bar{x}+1), \end{bmatrix} \tag{54b}$$

which has trace $\tau_J = -\gamma\bar{x}(\bar{x}+1)$, determinant $\Delta_J = \gamma^2(\beta - 2\bar{x} + 3\bar{x}^2)$, and eigenvalues $\lambda_{1,2} = (\tau_J \pm \sqrt{\tau_J^2 - 4\Delta_J})/2$.

## Takens-Bogdanov point

Noting that $\mathbf{J}$ has a double zero eigenvalue if $\tau_J = \Delta_J = 0$, we deduce that this occurs for $\alpha = \beta = 0$ at the fixed point $\bar{x} = 0$, about which the Taylor series expansion and normal form transformations of §1.1.3 are made. A nondegenerate sink also exists at $\bar{x} = 1$.

## Hopf bifurcations

When $\mathbf{J}$ has purely imaginary eigenvalues, or more specifically when $\tau_J = 0$ and $\Delta_J > 0$, a Hopf bifurcation can occur. Since $\gamma > 0$, $\tau_J = 0$ implies $\bar{x} = 0$ or $\bar{x} = -1$. The latter condition implies a closed glottis and so is biophysically irrelevant, and in the former case $\Delta_J = \beta$ must be positive. This yields the Hopf bifurcation set $\{\alpha = 0 | \beta > 0\}$: the vertical line emerging from the Takens-Bogdanov point in *Figure 10*. Crossing this set from left to right, the sink at $\bar{x} = 1$ becomes a source and a limit cycle appears or disappears.

The direction of bifurcation and stability of the limit cycles can be determined by computing the coefficient of a third order term in the normal form of the Hopf bifurcation, as described by *Guckenheimer and Holmes (1983)*, §3.4. At $\bar{x} = 0$ and $\alpha = 0$ the Jacobian (54) takes the form

$$\mathbf{J} = \begin{bmatrix} 0 & 1 \\ -\gamma^2\beta & 0 \end{bmatrix}, \tag{55}$$

with eigenvalues $\pm i\gamma\sqrt{\beta} \underset{=}{\mathrm{def}} \pm i\omega$. Using the similarity transformation

$$\begin{pmatrix} x \\ y \end{pmatrix} = \begin{bmatrix} 0 & 1 \\ \omega & 0 \end{bmatrix} \begin{pmatrix} u \\ v \end{pmatrix}, \tag{56}$$

*Equation (23)* becomes

$$\begin{pmatrix} \dot{u} \\ \dot{v} \end{pmatrix} = \begin{bmatrix} 0 & -\omega \\ \omega & 0 \end{bmatrix} \begin{pmatrix} u \\ v \end{pmatrix} + \begin{pmatrix} \frac{\gamma^2 v^2}{\omega} - \gamma u v - \frac{\gamma^2 v^3}{\omega} - \gamma u^2 v \\ 0 \end{pmatrix}, \tag{57}$$

and we may use Equation (3.4.10) of *Guckenheimer and Holmes (1983)* to determine that

$$a = -\frac{\gamma}{8}\left(1 + \frac{1}{\beta}\right) < 0, \tag{58}$$

implying that the limit cycles are stable and lie to the right of the bifurcation set $\{\alpha = 0 | \beta > 0\}$, in region I or region III (depending on the value of $\beta$).

## Saddle-node and pitchfork bifurcations

In 2-dimensional systems like *Equation (23)* a pair of fixed points, either a saddle and a source or a saddle and a sink, come together in a degenerate saddle-node bifurcation and disappear (or, crossing the bifurcation set in the opposite direction, appear and separate). A saddle node occurs when one of the eigenvalues of $\mathbf{J}$ is zero and the other is not, i.e., for $\Delta_J = 0$ with $\tau_J \neq 0$. Alternatively, they may be located by seeking double roots of *Equation (53)*, which occur when its first derivative also vanishes

$$3\bar{x}^2 - 2\bar{x} + \beta = 0. \tag{59}$$

Substituting the solution $\bar{x} = (1 \pm \sqrt{1 - 3\beta})/3$ of (59) into *Equation (53)* and noting that $\bar{x}^2 = (2 - 3\beta \pm 2\sqrt{1 - 3\beta})/9$, we obtain the two saddle-node bifurcation curves:

$$\alpha = \frac{2 - 9\beta \pm 2(1 - 3\beta)^{\frac{3}{2}}}{27}, \quad \text{with} \quad \beta < \frac{1}{3}. \tag{60}$$

As shown in *Figure 10*, these curves meet in a cusp at the pitchfork bifurcation point $(\alpha, \beta) = (-1/27, 1/3)$, at which the unique fixed point $\bar{x} = 1/3$ is a triple root of *Equation (53)*. Note that the Takens-Bogdanov point lies on the left hand curve, where the Hopf bifurcation line begins.

## Homoclinic loop bifucations

Unfolding the Takens-Bogdanov normal form as in *Guckenheimer and Holmes (1983)*, §7.3 shows that a third bifurcation curve emerges from the point $(\alpha, \beta) = (0, 0)$ to the right of $\alpha = 0$ and tangent to it at $(0, 0)$. On this curve, shown dashed in *Figure 10*, one of the separatrices emerging from the saddle point returns to it in a homoclinic loop. Approaching from above and to the left in region III the stable limit cycle grows until, for parameters $(\alpha, \beta)$ on this homoclinic bifurcation curve, it fuses with the saddle and disappears. Numerical solutions indicate that the curve crosses regions III and II, thereby dividing them, and ends on the upper saddle-node bifurcation curve. To the right of this meeting point, the saddle-node bifurcation occurs on a closed cycle, creating finite amplitude limit cycles whose periods rapidly decrease from infinity moving upward and rightward into region I. See (*Perl et al., 2011*, *Figure 4*), but note that a different parameterization is used in that paper. The extreme sensitivity of the period and waveform of these oscillations are largely responsible for the broad spectral content and 'uncontrollable' aspects of infant cries produced by the model.

No general methods exist for finding homoclinic bifurcation curves analytically, although they can be approximated in the neighborhood of multiply-degenerate fixed points such as the Takens-Bogdanov point. In fact a bifurcation set topologically equivalent to the present one has been found in a version of the forced van der Pol oscillator (*Holmes and Rand, 1978*), cf. *Guckenheimer and Holmes (1983*, §2.1).

