## [Decision Letter]

Thank you for submitting your article "Vocal development in an integrated biological landscape" for consideration by *eLife*. Your article has been reviewed by two peer reviewers, and the evaluation has been overseen by Reviewing Editor David Kleinfeld and Timothy Behrens as the Senior Editor. The following individual involved in review of your submission has agreed to reveal their identity: David Golomb (Reviewer #1).

The reviewers have discussed the reviews with one another and the Reviewing Editor has drafted this decision to help you prepare a revised submission.

Summary:

The development of vocalization proceeds at many physiological levels and includes changes in the geometry of the vocal tract, changes in the strength of respiratory and laryngeal muscles, changes in neuronal drive, and changes in the nature of feedback based on maturity and social interactions. The authors use computational and theoretical methods to model this process for the development of marmoset calls. A key realization is the need for a hierarchical model that progressively changes one aspect of the vocal system, e.g., geometry of the tract, until the desired target call remains out of bounds, at which point changes in a different set of physical parameters, e.g., muscular strength, occur. This approach is conjectured to mimic the changes that occur in development under evolutionary pressures.

Essential revisions:

All reviewers and the Reviewing Editor appreciate the importance of your approach. Reviewer #1 has some straightforward queries about the mathematics. However, the less mathematically savvy reviewer, #2, had difficulty in understanding your procedure. We encourage you to carefully edit the manuscript. I would suggest that the PI have neuroethology colleagues read the manuscript before resubmission. Further, the Reviewing Editor feels that terms need to be thoroughly explained, e.g., be explicit about the notional of a landscape in terms of the optimization procedures, spell out the machinations of Jayne's softmax procedure, etc.

Please address all of the issues in the two reviews below. Please heed the call of the second reviewer – a well-known expert in animal vocalization and someone that you would wish to comprehend and espouse on your work.

*Reviewer #1:*

The authors combine theoretical, computational and experimental methods to explain how the sound made by marmoset monkeys shifts from immature "cry" to mature "phee". A biomechanical model of voice production is described in "Material and methods". In contrast to the usual structure of research paper, its analysis is carried in that section. The derivation of the mechanical model, based on normal form transformation, is explained in the Appendix. The modification of model parameters, from those that lead to "cry" to those that lead to "phee", is described in the Results section. It is assumed that the two parameters, air pressure (α) and vocal fold tension (β) are equal and have both a value theta. A cost function of theta is defined, and several variations of this cost function, each one with more terms than the previous one, are examined. The most elaborated cost function accounts for the biomechanical, muscle and neural factor but not for the social factor.

The major experimental results and a brief description of the model appeared in a previous publication by the same group (Takahashi et al., Science, 2015). The derivation of the model using normal form analysis and the material about the cost function are new and justify a new publication where a simple model is used to explain development of animal behavior. I like the fact that the authors report what the model cannot explain changes in amount of parental feedback. There are several points that need rewriting or clarification.

1) Subsection 4.12, "Properties of the biomechanical model" describes results, and should therefore be moved the Results section, together with Figure 9. This inclusion will make the Results section more coherent more clear to read.

2) α and β are used as bifurcation parameters. In reality, they vary with time (for example, in Fig, 8). What is the dynamics of α and β during vocal activity (not the long-time scale of development)?

3) The authors assume that α=β=theta, and claim that "other choices of α and β that include the three types of calls yield similar results (subsection “2.3 Development of the vocal tract”). In contrast, Figure 9 shows that when α is reduced, the system undergoes a Hopf bifurcation (for large enough β), whereas when β is reduced, the system always undergoes a saddle-node bifurcation that yields slow limit cycles. Therefore, I'd expect that varying α has different effects than varying β.

4) I do not understand the role of the grey lines representing eta in Figure 3.

*Reviewer #2:*

When I received the manuscript "Vocal development in an integrated biological landscape", by Teramoto et al., I thought I would be capable of writing a report. After reading the manuscript in detail, I am afraid I can only evaluate parts of the material.

The manuscript deals with the development of vocalizations in marmosets, and starts with a generic statement which is difficult to disagree with: "Vocal development is the adaptive coordination of the vocal apparatus, muscles, respiration and the nervous system". The authors then propose a way to translate Waddington's landscale metaphor into a quantitative way to address this developmental question.

My main problem with the manuscript is the abysmal difference between the sound, solid and rigorous treatment of the dynamics of the vocal apparatus, with the highly metaphoric ways in which the physiological instructions are treated. I have no way to evaluate these last parts of the work.

In my opinion, the "softmax action selection rule" as a way to predict the probability of uttering a cry, subharmonic sound or a phee call is at the antipodes of the detailed way in which the normal form of the equations for the vocal organ dynamics' is treated, and I have a hard time in putting all these abysmally different strategies together.

If a partial evaluation is of any use, the analysis of the vocal fold dynamics is excellent and sound. And of course, the general idea of the work, with its intention to cover the problem in an integrated way, intellectually appealing.

---

## [Author Response]

*Essential revisions:*

*All reviewers and the Reviewing Editor appreciate the importance of your approach. Reviewer #1 has some straightforward queries about the mathematics. However, the less mathematically savvy reviewer, #2, had difficulty in understanding your procedure. We encourage you to carefully edit the manuscript. I would suggest that the PI have neuroethology colleagues read the manuscript before resubmission. Further, the Reviewing Editor feels that terms need to be thoroughly explained, e.g., be explicit about the notional of a landscape in terms of the optimization procedures, spell out the machinations of Jayne's softmax procedure, etc.*

*Please address all of the issues in the two reviews below. Please heed the call of the second reviewer – a well-known expert in animal vocalization and someone that you would wish to comprehend and espouse on your work.*

In the revised version of the manuscript, we have addressed all of reviewer 1’s queries. We also took very seriously the concerns of reviewer 2. We completely agree that if experts in the biology of animal communication cannot understand our manuscript, then we failed to achieve our primary goal. As was suggested, we had an expert in vocal communication (Morgan Gustison, University of Michigan) read our original submission. She expressed similar concerns to those raised by reviewer 2 and provided an enormous number of useful suggestions to enhance comprehension. As a result, in the last two months, we have completely rewritten the manuscript to make more clear our approach and findings. New figures have also been added to facilitate comprehension.

We also received some brief comments from a developmental psychologist to whom we sent the revised manuscript.

David Lewkowicz (Northeastern University) wrote: “I did have a quick look at it and am very impressed with the scope and conceptualization of the work. It seems to be a beautiful developmental systems analysis of a key component of behavior in the context of a rigorous mathematical model with a really masterful visualization of the phenomenon in the accompanying video. The caveats section is very important and makes the paper much stronger. I think that it is a key section because, as you well know, these sorts of models are only as good as the number of parameters you include in them and the theoretical assumptions underlying their choice. It does seem, though, that your model can nicely account for the changes in the nature of the vocalizations… Congrats on a nice piece of work.”

We are hopeful that this version of the manuscript will be able to reach a much broader audience, especially those interested in animal communication, its biology and development.

*Reviewer #1:*

*[…] 1) Subsection 4.12, "Properties of the biomechanical model" describes results, and should therefore be moved the Results section, together with Figure 9. This inclusion will make the Results section more coherent more clear to read.*

Thank you for this suggestion. We understand the point of view of the reviewer and we initially moved subsection 4.12 up to Results section in many iterations of the revised manuscript. Nevertheless, after the feedback we received from our ethologist colleague as well as our own multiple readings of the revised manuscript, we felt that subsection 4.12 is quite technical/mathematically-sophisticated and not really necessary for a less mathematically savvy reader to understand the rest of the Results. Therefore, with the aim of being more comprehendible for a broader readership, we moved back the subsection 4.12 to the Materials and methods. In an effort to give enough information for the more mathematically inclined reader, in this revised version of the manuscript, we cite the appropriate equations and sections of the Materials and methods and the Appendix so that the reader can follow the mathematical details if they so desire.

*2) α and β are used as bifurcation parameters. In reality, they vary with time (for example, in Fig, 8). What is the dynamics of α and β during vocal activity (not the long-time scale of development)?*

Figure 9 (formerly Figure 8) shows phase portraits and time histories of two calls with different but constant values of α (air pressure) and β (vocal fold tension). We make clear this fact in the legend for the figure in the revised manuscript.

The only place where the effects of changing α and β appear are in the lower panels of Figure 3 of the revised manuscript. This shows the transition calls produced by the model. We have inserted a sentence explaining the dynamics of α and β for those cases in section 3.1, second paragraph.

*3) The authors assume that α=β=theta, and claim that "other choices of α and β that include the three types of calls yield similar results (subsection “2.3 Development of the vocal tract”). In contrast, Figure 9 shows that when α is reduced, the system undergoes a Hopf bifurcation (for large enough β), whereas when β is reduced, the system always undergoes a saddle-node bifurcation that yields slow limit cycles. Therefore, I'd expect that varying α has different effects than varying β.*

We agree that α and β have different effects on calls, as summarized in

Figure 3 of the revised manuscript. These figures show that the effect of α is primarily to change call amplitude, while simultaneous change in α and β change call frequency. Since the transition from infant cries and subharmonics to adult phees is primarily expressed in call frequency, and the slow limit cycles corresponding to broad-band cries occur only above and near the upper saddle-node bifurcation curve (Figure 10, formerly Figure 9), we chose to focus on the diagonal path theta = α = β. In evaluating the cost function and associated probabilities, this leaves the bifurcation curve transversely. This path crosses regions in which the model produces all the calls of interest (see Figure 3). To clarify this, we have added the theta axis to the bifurcation set in Figure 10 and noted its use in the legend.

*4) I do not understand the role of the grey lines representing eta in Figure 3.*

The grey lines show the phee/cry ratios predicted by the model for different values of eta, higher values of which concentrate the probability distribution in the phee call region (see Figure 3, now Figure 4). Our conclusion is that the cost function –log g(theta) cannot predict the phee/cry ratio population data, which transitions from cries to phees between 10 and 40 postnatal days. The detailed explanation can be found in the final paragraph in section 3.2 (revised manuscript) and in the figure legend.

*Reviewer #2:*

*When I received the manuscript "Vocal development in an integrated biological landscape", by Teramoto et al., I thought I would be capable of writing a report. After reading the manuscript in detail, I am afraid I can only evaluate parts of the material.*

*The manuscript deals with the development of vocalizations in marmosets, and starts with a generic statement which is difficult to disagree with: "Vocal development is the adaptive coordination of the vocal apparatus, muscles, respiration and the nervous system". The authors then propose a way to translate Waddington's landscale metaphor into a quantitative way to address this developmental question.*

*My main problem with the manuscript is the abysmal difference between the sound, solid and rigorous treatment of the dynamics of the vocal apparatus, with the highly metaphoric ways in which the physiological instructions are treated. I have no way to evaluate these last parts of the work.*

*In my opinion, the "softmax action selection rule" as a way to predict the probability of uttering a cry, subharmonic sound or a phee call is at the antipodes of the detailed way in which the normal form of the equations for the vocal organ dynamics' is treated, and I have a hard time in putting all these abysmally different strategies together.*

*If a partial evaluation is of any use, the analysis of the vocal fold dynamics is excellent and sound. And of course, the general idea of the work, with its intention to cover the problem in an integrated way, intellectually appealing.*

We thank the reviewer for appreciating our integrative approach to vocal development. We are chagrined that our writing and presentation lacked clarity and that, as a result, the reviewer felt that there was an abyss between our biomechanical model and the softmax action selection. Both are, in fact, necessary components of the optimal control approach that serves as the foundation of our study. Nothing we did methodologically or analytically is a departure from the standard and well-established application of optimal control theory for motor learning in other domains. But we did fail to explain things clearly.

In this revised manuscript, we went to great lengths to improve the writing and make the concepts and strategies as clear as possible. We asked an expert in animal communication to carefully read our manuscript, and based on the feedback from her and our own rethinking and rereading of multiple revised drafts, we made large-scale changes to the manuscript. In particular, we hope the reviewer will notice that all sections – the Abstract, Introduction, Results, Discussion, and Methods – were rewritten to better explain each of the concepts used in the manuscript. New figures have also been added to enhance comprehension. Here we highlight some of the many changes we made:

1) The Abstract now more clearly states the goals, approach and findings of our study.

2) The last paragraph of the Introduction now clearly lays out the three behavioural phenomena we seek to explain with the landscape. We’ve also added figure panels Figure 1 to illustrate what we are investigating. These three phenomena are referred back to throughout the Results text.

3) The “Overview of Approach” is now separated from the Results and we elaborate more about our inferential process in this section. Moreover, we added Figure 2 to illustrate the sequence of steps we used when applying optimal control theory to vocal development.

4) Following the presentation of the biomechanical model, we better motivate each section of the Results as a biological query as opposed to a “computational model” query.

5) The Discussion now more clearly addresses our findings, our approach and its limitations followed by a discussion of the potential applications of our vocal development landscape.

6) We also added a video that includes an animation of an individual’s vocal development in the Waddington-like landscape.

We really appreciate reviewer 2’s direct feedback on our original manuscript. We hope that the revised manuscript is not only more comprehensible but perhaps even engaging to read by our target audience: biologists studying animal communication.